# A fully automated benchmarking suite to compare macromolecular complexes

Gabriel Studer [1,2,3], Xavier Robin [1,2,3], Stefan Bienert [1,2], Janani Durairaj [1,2], Peter Škrinjar [1,2], Gerardo Tauriello [1,2], Andrew Mark Waterhouse [1,2] & Torsten Schwede [1,2] ✉

Protein structure prediction has a long history of benchmarking efforts such as critical assessment of structure prediction, continuous automated model evaluation and critical assessment of prediction of interactions. With the rise of artificial intelligence-based methods for prediction of macromolecular complexes, benchmarking with large datasets and robust, unsupervised scores to compare predictions against a reference has become essential. Also, the increasing size and complexity of experimentally determined reference structures by crystallography or cryogenic electron microscopy poses challenges for structure comparison methods. Here we review the current state of the art in scoring methodologies, identify existing limitations and present more suitable approaches for scoring of tertiary and quaternary structures, protein–protein interfaces and protein–ligand complexes. Our methods are designed to scale efficiently, enabling the assessment of large, complex systems. All developments are available in the structure benchmarking framework of OpenStructure. OpenStructure is open source software and available for free at https://openstructure.org/.

The field of protein structure modeling has a long history in benchmarking the accuracy of prediction methods. Various numerical scores are used to systematically compare the computationally generated models against experimental ground truth, reference structures (also known as 'target' or 'gold standard'). The double-blind critical assessment of structure prediction (CASP) experiment[1] has biennially assessed state-of-the-art methodologies since 1994[2] and is a major driver advancing the field. The complementary continuous automated model evaluation (CAMEO)[3] project assesses automated servers every week and is driving the development of fully automated prediction and assessment methods. The critical assessment of prediction of interactions (CAPRI) experiment[4,5] assesses the accuracy of the interface prediction in protein–protein complexes, with new rounds announced approximately every 6 months. CASP and CAPRI have been collaborating since 2016[6]. A variety of ligand pose prediction challenges have taken place, such as D3R[7–10], CELPP[11] and others[12–16], and ligand assessment has been included in CASP since 2022[17]. Over the past decades, objective, blind, independent benchmarking efforts have been a major driver for the development of highly accurate structure prediction methods[18]. However, despite the impressive recent progress in prediction accuracy, even the latest developments in the field of structure prediction such as AlphaFold[19,20] and RoseTTAFold[21] have their limitations, highlighting the importance of continuing benchmarking efforts in the future. Large datasets[22] and robust scores that can be applied in a high-throughput manner without human intervention are essential for the development of data-driven artificial intelligence (AI)-based prediction methods.

## Chemical mapping

To allow for a comparison between a prediction and the reference structure, a one-to-one mapping between all equivalent chemical molecules (polymer chains, small molecule ligands, etc.), in the reference structure and in the model must be established. For robust benchmarking, the chain mapping problem is defined as the task of establishing a one-to-one assignment between chains in the model and the reference structure, such that the mapping is optimal with respect to the scoring metric used to evaluate model quality. This ensures that benchmarking results reflect the best possible structural correspondence rather than artifacts of arbitrary chain assignments.

[1]Biozentrum, University of Basel, Basel, Switzerland. [2]SIB Swiss Institute of Bioinformatics, Basel, Switzerland. [3]These authors contributed equally: Gabriel Studer, Xavier Robin. ✉e-mail: torsten.schwede@unibas.ch

For polymers, this 'chain mapping' is only trivial in cases of monomeric predictions. Extending this mapping to complexes with multiple polymer chains has long been a challenge. The naive approach consists of enumerating all possible mappings, computing a score and selecting the optimal result. This quickly becomes intractable as the complexity of the problem grows factorially with the number of chains. Historically, several simplifications of this problem have been introduced to allow the scoring of complexes. The CAPRI community[4,5] developed interface-centric scores solely applicable to dimeric complexes, bypassing the need for global pairwise mappings. Only recently, CAPRI has started to assess larger assemblies by averaging per-interface scores[4].

Most current tools aim to derive an explicit one-to-one mapping by heuristically optimizing for rigid superposition based scores. Notably, the MM-align tool[23] aims to maximize the template modeling score (TM-score)[24] between model and reference by enumerating the full solution space but omits the costly score computation for unlikely mappings. US-align[25] also maximizes the TM-score, but avoids exhaustive enumeration by deriving an initial mapping with a greedy search algorithm, which is then iteratively optimized. The algorithm described for AlphaFold-Multimer[26] selects an anchor chain in the reference and initializes mappings by superposing all matching-sequence model chains followed by iterative chain pairing by minimal centroid distance, searching for the mapping with lowest centroid root mean squared deviation (r.m.s.d.). Foldseek-Multimer[27] performs all-versus-all superpositions and clusters transformation matrices to identify compatible chain sets for mapping. For non-superposition-based scores, the problem can sometimes be reduced to the identification of mappings between symmetry related groups[28]. To date, chain mapping methods specifically designed for these types of scores remain absent in the field.

Small-molecule ligands can be matched with graph-based methods[29,30]. Here, we refer to it as 'ligand assignment', to distinguish it from the polymer 'chain mapping' term. Challenges are similar to those experienced for polymers, with the additional consideration of symmetrical groups within ligands such as phenyl groups, where atoms cannot be unambiguously assigned.

Here, we introduce a consistent framework to establish mappings between any number of compounds in a reference structure, be it protein, DNA, RNA or small molecule ligand, and their counterpart in the predicted model. This flexible framework allows us to compute a large array of scores to assess different aspects of the quality of predicted macromolecular complexes.

## Comparison scores

In this context, we use the term 'score' specifically to refer to benchmarking metrics that quantify the agreement between a predicted model and a reference structure. This usage is distinct from scores that may reflect energy-based evaluations, such as those generated by tools such as ZRank[31], which are used during modeling or docking but are not direct measures of structural similarity.

Tertiary structure scores can be broadly categorized into two groups. First, scores reliant on rigid superposition of representative backbone atoms (typically Cα for proteins), such as the r.m.s.d.[32], global distance test (GDT)[33] or TM-score[24]. The r.m.s.d. has been largely abandoned in this context owing to its sensitivity to outliers and movements of individual protein domains. In addition, the r.m.s.d. requires subsets of mapped atom positions, meaning it does not penalize for missing residues in incomplete models and ignores any extra atoms present in one structure that are not found in the other. CASP mitigates for the effects of domain movements to some extent by manually segmenting reference structures into rigid substructures for separate evaluation. However, structural flexibility remains a challenge for fully automated benchmarking initiatives, such as CAMEO, and large-scale comparisons required for data-intensive applications in the field of AI. Here, a second group of scores plays a crucial role. Local and superposition independent scores are less sensitive to domain movements by focusing on

differences in the local environment[34]. Examples include the contact area difference (CAD) score[35], and the local distance difference test (LDDT) score[36]. Both scores consider all heavy atoms and thus require correct sidechain placement to achieve optimal values.

Benchmarking protein assemblies requires a set of specialized scores to focus on accuracy of the interfaces. CAPRI roughly classifies the similarity of a prediction to the reference structure as 'incorrect', 'acceptable', 'medium' or 'high' on the basis of ligand r.m.s.d.(L-RMSD), interface r.m.s.d. (i-RMSD) and $f_{nat}$ (ref. [4]). The DockQ score[37] was introduced in CAPRI recently as an effort to combine these three scores into one continuous number, avoiding a classification approach and thus making it more suitable as a target score for automated modeling methods[38]. The CASP assessment of oligomers primarily relies on interface contact-based scores, akin to $f_{nat}$, named the interface contact similarity (ICS) and interface patch similarity (IPS) scores[6]. To encompass the accuracy of the individual subunits and overall topology, these interface-centric scores have been supplemented by scores originally devised for tertiary structure comparison, including LDDT and TM-score[39–41]. When the modeling challenge includes predicting the stoichiometry, as in CAMEO, the QS-score[28] is appropriate as it discriminates between alternative quaternary structures and different stoichiometries (Supplementary Section 3).

Predicting how a small-molecule ligand binds to a protein target, also known as pose prediction, is an important task in drug discovery. Previous ligand pose prediction challenges employed two main types of scores to assess how well participants could model receptor–ligand complexes—a symmetry-corrected r.m.s.d. to measure the absolute accuracy of the predicted ligand within the binding site—and contact-based scores to evaluate the reproduction of native receptor–ligand noncovalent interactions[13,42,43]. The GPCR dock[15,16] and the first stage of the D3R Grand Challenge 3[9] also challenged their participants to model the conformation of the receptor protein. The assessment required an additional superposition of the model onto the reference structure.

CASP15 presented a more complex challenge. Participants were tasked to model entire protein–ligand complexes including the receptor, sometimes as an oligomer, and potentially multiple ligands. Preexisting methods were unable to score these complex predictions out of the box. Spyrmsd[30] computes symmetry-corrected r.m.s.d. for a single reference model ligand pair in the same frame of reference, doesn't include binding site detection and superposition, and does not generalize to complexes containing several ligands. Similarly, previous fingerprint-based scores assessing protein–ligand interactions[13,42–44] are restricted to a subjective set of interactions (such as hydrogen bonds, ionic, hydrophobic or π interactions, or metal coordination) and dependent on manual preparation steps, making them difficult to reproduce consistently. Therefore, new ligand assessment methods with automated ligand identification, chain mapping for oligomers and superposition of the receptor had to be developed[29]. The resulting scores, binding site superposed symmetry-corrected r.m.s.d. (BiSyRMSD) and LDDT-protein–ligand interactions (LDDT-PLI), have since been refined and their implementation is described in detail in this paper.

## Aim of this manuscript

We describe a fully automated, fast and reliable suite of tools to compare theoretical models with experimental reference structures, implemented in the OpenStructure structural biology framework. We discuss strengths and limitations of the various scores, and offer recommendations to guide researchers on aspects that require special attention. OpenStructure automatically applies the necessary steps to compute the scores, including state-of-the-art algorithms for chain mapping. OpenStructure provides a large array of complementary scores to assess the accuracy of different types of predictions (summarized in Fig. 1) including protein, DNA or RNA tertiary structures; single or

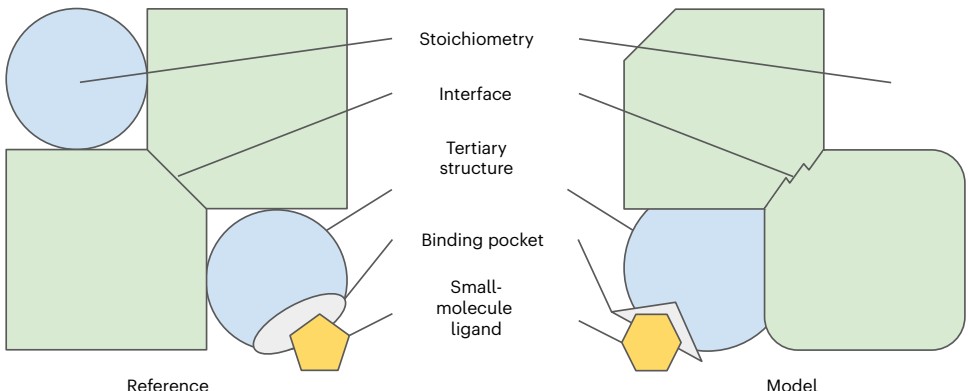

**Fig. 1 | Different aspects of structure prediction assessment.** Schematic example of a hetero-2-2 dimer reference with a small-molecule ligand (left). Evaluating a model (right) includes mapping equivalent components and scoring differences to the reference. Inaccuracies of the model may include incorrect stoichiometry of the complex, structural differences in the tertiary structures and in interfaces, incorrect size and shape of the binding site, and wrong conformation or pose of the small-molecule ligand. In contrast to the 'spot seven differences' game, differences must be assessed not just qualitatively but quantitatively.

multiple polymer–polymer interfaces; small molecule and ion poses and geometry of the ligand binding pocket; and the stoichiometry of predicted macromolecular complexes. Together, these scores paint a comprehensive view of how accurately a model represents the target structure[45]. Providing all these scores in a single, consistent and highly optimized framework greatly facilitates high-throughput benchmarking experiments, for both method developers and benchmark operators. The OpenStructure scoring framework was used in recent CASP and CAMEO experiments.

## Results

### Mapping polymer chains

To date, the field has lacked a method adept at deriving chain mappings for large assemblies, while operating on superposition-independent scores to mitigate the limitations of superposition-dependent methods. In this work, we present QSMap—a heuristic algorithm that optimizes for the interface contact-based QS-score or backbone LDDT (Cα for protein and C3′ for nucleotides). For applications where overall topology is of relevance, we provide QSMapR, which optimizes for backbone r.m.s.d. (Cα for peptides and C3′ for nucleotides). Like the chain mapping algorithm utilized by AlphaFold-Multimer, QSMap/QSMapR are sequence dependent, that is, reference/model chains are mapped within groups that are considered chemically equivalent. This is a desired property in a benchmarking scenario. All approaches are described in detail in the Methods section.

Two test datasets were constructed to assess the practical limits for QSMap/QSMapR, and to compare their performance on a real world benchmarking scenario. In addition, we compared QSMapR with US-align[25], Foldseek-Multimer[27] and our own implementation of the chain-mapping algorithm used in AlphaFold-Multimer[26], all of which rely on global superposition and are optimized for assessing overall topology. The first set consists of a maximum of ten randomly selected structures retrieved using the RCSB PDB search API[46] (see Data availability statement) for homomers with increasing numbers of chains, $N$. For cases where fewer than ten structures were available, we performed data augmentation by randomly selecting larger experimental structures and truncating them to contain only the first $N$ chains. Mappings have been performed using these oligomers as both model and reference structures, performing a chain mapping on the structures themselves. The second set consists of models generated for the CASP15 assembly modeling challenge[41] excluding trivial cases, such as dimers and hetero-oligomers with one-to-one chain assignment. The dataset comprises 3,559 models of varying stoichiometry, ranging from homo-trimers to hetero 9-9-9-mers.

In general, QSMap/QSMapR can handle problem sizes involving up to 10 polymer chains with runtimes in the order of seconds or 30 chains in the order of 100s of seconds (a single thread of an AMD EPYC 7742 processor). As the number of chains increases beyond this threshold, runtimes gradually increase and become impractical. This is substantially better than the approximately ten chains that are tractable by naive enumeration (Fig. 2a). QSMap chain mappings outperform QSMapR chain mappings in contact-based comparisons, as indicated by QS-score (Fig. 2b) and other similar scores, including ICS and LDDT (Supplementary Fig. 1). For comparisons focusing on overall topology, that is, rigid superposition-based comparisons such as TM-score, chain mappings from QSMapR perform better (Fig. 2c). In essence, each algorithm excels in the specific aspects they optimize for. In terms of runtime, QSMap/QSMapR successfully establish a chain mapping for all test cases, rarely exceeding 100 s (Supplementary Fig. 4a). QSMapR produces chain mappings that are superior to Foldseek-Multimer and AlphaFold-Multimer and as accurate as US-align but approximately one order of magnitude faster (Supplementary Section 1). To conclude, QSMap is recommended for contact-based scenarios, while QSMapR should be preferred when overall topology is the primary concern.

### Updated LDDT reference implementation

The LDDT measures differences in distance between every atom pair within a defined inclusion radius, henceforth termed as 'contact'. It was introduced in the CASP9 experiment[47] and has been used as the primary comparison score for tertiary structures in CAMEO as it allows for fully automated assessment owing to its robustness against domain movement events. The score applies stereochemistry checks to penalize for serious stereochemical irregularities and was originally restricted to single-chain proteins. Starting with CASP13[39] and CAMEO[3], LDDT was extended to evaluate protein quaternary structures. However, already at CASP13, the employed chain mapping algorithm proved insufficient to process large assemblies and needed input from external tools in these cases[39]. With the shifting focus of the modeling community to macromolecular complexes[3] and interest in applying the concept of superposition-independent distance differences to RNA or small molecules, we introduce a new LDDT reference implementation.

This implementation successfully processes large assemblies by tightly integrating with the QSMap chain mapping algorithm (see the QSMap section in Methods), and was extended to nucleotides. In addition, two variations have been added: (1) i-LDDT, which considers only distances across interfaces and (2) bb-LDDT, which considers only representative backbone coordinates (Cα for peptides and C3′ for nucleotides). Extensive testing and comparisons with other scores were conducted in

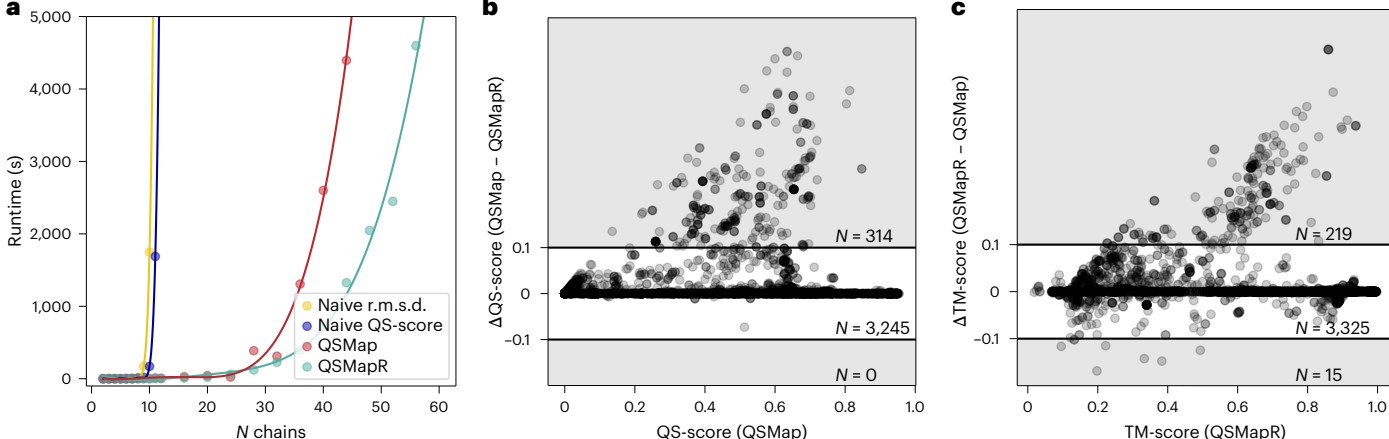

**Fig. 2 | QSMap/QSMapR benchmark. a**, The average runtimes for homo *N*-mers to determine tractable problem sizes. Naive enumerations are impractical for *N* > 10, whereas QSMap and QSMapR enable mapping of larger structures. **b**, QSMap often finds chain mappings with higher QS-scores compared with QSMapR. **c**, For comparisons based on global superposition-based metrics, QSMapR is the preferred method. **b**,**c**, Shaded areas highlight data points with notable score differences (absolute difference ≥ 0.1).

the context of the CASP15 RNA modeling challenge[48]. Stereochemistry checks for nucleotides were not implemented for CASP15 but have been implemented in the context of this manuscript and CASP16.

## Small-molecule ligands

To assess the accuracy of predictions of small-molecule ligands in a complex with a target structure, such as cofactors, inhibitors or drug compounds, in the context of CASP15[29], we developed three new scores. The BiSyRMSD is a symmetry-corrected r.m.s.d. in Å, which measures the absolute accuracy of a ligand pose prediction computed after superposing the binding site coordinates of the model onto the reference. The binding site is defined as any residue with at least one atom within 4 Å of the ligand, excluding hydrogen atoms, based solely on the reference structure. A local superposition was needed as the reference structures in CASP were larger and more flexible than those in GPCR dock[15,16] and the first stage of the D3R Grand Challenge 3[9]. Lower values indicate more accurate predictions, with scores below 2 Å typically interpreted as 'success' in docking experiments. While there is no strict upper limit to the score, ligands posed more than 20 Å away from their correct position might cause the binding site detection to fail if they are positioned far away from the chain they interact with. To mitigate limitations of r.m.s.d.-based scores of incomplete predictions, we require the entirety of the reference ligands to be modeled.

LDDT-PLI is an LDDT score restricted to polymer–ligand atomic contacts, which assesses the reproduction of native contacts by looking at every atom pair within a defined inclusion radius, and penalizes contact overprediction in the model. Like LDDT, LDDT-PLI is constrained between 0 and 1, with higher values indicating more accurate results. A non-zero LDDT-PLI score indicates that the ligand was modeled in the right pocket, and quickly goes down to 0 as contacts become unfulfilled. However LDDT-PLI can remain higher than 0 even with large BiSyRMSD values if the part of a flexible ligand making contact with the polymer is modeled accurately. Finally, LDDT-ligand pocket (LDDT-LP) is an LDDT restricted to atomic contacts between polymer residues of the binding site, and is constrained between 0 and 1. The value of LDDT-LP is 0 when the binding site consists of a single residue mapped between the model and the reference. All scores take care of chain mapping, symmetry correction for ligands and, when multiple ligands are present, generate a ligand assignment where no reference or model ligands can be part of more than one PLI (for details, see Methods). Here, we discuss some properties of these scores in more detail.

In order to investigate the scores' behavior, we gathered all models that were assessed in the CASP15 ligand modeling challenge[29]. Figure 3a

shows the relationship between LDDT-PLI and BiSyRMSD (plotted on a log scale). Missing values (when scores were missing or could not be computed; see 'Ligand assignment' section in the Methods) are marked with a triangle. While the two scores are strongly negatively correlated (Spearman $\rho$ = −0.989), a few interesting outliers where the two scores deviate from the correlation line can be observed. Figure 3b shows an example where a BiSyRMSD of 0.02 Å indicates a spot-on prediction, with a very low backbone superposition r.m.s.d. of the binding site of 0.37 Å. However, some side chains in the binding site are flipped (Asp72 and Asn77), which results in slightly lower LDDT-LP (0.83) and LDDT-PLI (0.902). Another source of discrepancy is shown in Fig. 3c, where a part of the ligand was modeled in the correct binding pocket (Fig. 3c, left), resulting in a non-zero LDDT-PLI of 0.26, but a disconnected part of the ligand was modeled more than 100 Å away (Fig. 3c, right). As expected, the BiSyRMSD applies a square penalty to these very far atoms, resulting in a score of 87.5 Å. A third example is shown in Fig. 3d, where a magnesium atom is placed 0.67 Å away from the correct position. However, because the ligand is still located at the same correct distance from the atoms of the binding site residues, there is almost no penalty to LDDT-PLI (0.99).

Figure 4 illustrates the effect of the extra penalty in LDDT-PLI for the additional contacts in the model. In Fig. 4a, an additional chain (in pink) was modeled to pass through the binding site and clashes both with the ligand and the binding site. Without this chain, the model would result in almost perfect scores (BiSyRMSD <0.5; LDDT-LP and LDDT-PLI both >0.9). However, with the penalty for added contacts, LDDT-PLI becomes 0.53, indicating an average prediction accuracy. Figure 4b shows a more subtle case of loop and side chain misplacement. The terminal loop, and in particular Arg6, is modeled closer to the ligand than in the reference. This results in an LDDT-PLI of 0.65, while BiSyRMSD and, to a lesser extent, LDDT-LP do not suffer from this as much, with scores of 1.11 and 0.80, respectively. While effective at detecting deviations from the reference, the extra added contacts penalty should not replace stereochemical checks on the model.

A limitation of the ligand scores is that they are restricted to interactions between polymer chains (proteins or nucleic acids) and small-molecule ligands by definition, and do not consider other small molecules or short peptides the ligand might be interacting with. This can be an issue for ions interacting with organic molecules rather than with the protein, such as in the CASP target T1118v1 where iron atoms interact with macrocyclic bisucaberin ligands, but not with the FoxA protein. As a result, no score can be computed for the iron atoms with the default parameters. A workaround is to increase the binding site

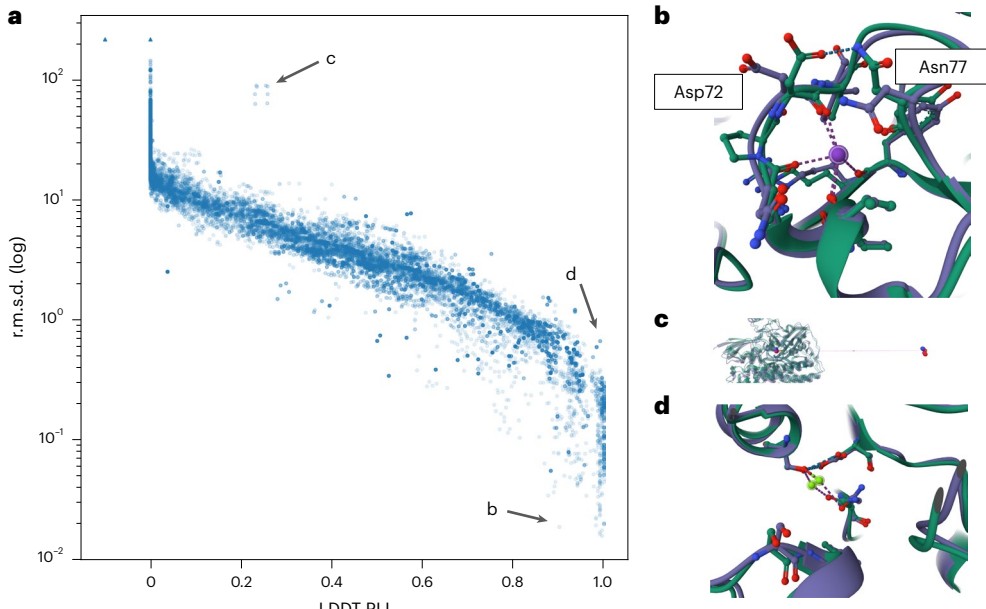

**Fig. 3 | Relationship between BiSyRMSD and LDDT-PLI. a**, A scatter plot of LDDT-PLI (*x* axis) and BiSyRMSD (*y* axis, log scale). Each point (*n* = 31,905) represents an assigned reference ligand pair. Pairs for which LDDT-PLI (*n* = 7,798) or BiSyRMSD (*n* = 7,921) could not be computed are represented with a triangle and values were replaced with −0.1 (LDDT-PLI) or a value of 1.5 times the highest BiSyRMSD score observed in the analysis. Points have 10% opacity, and more solid points indicate multiple identical predictions. Interesting outliers are marked with arrows. **b–d**, Examples of ligand outlier pose predictions with disagreeing LDDT-PLI and BiSyRMSD models highlighted in **a**, showing flipped side chains (**b**), a disconnected ligand (**c**) and misplaced ion conserving inter-atomic distances (**d**). Model chains are depicted in purple and reference chains in green. Binding site residues and ligands are shown as balls and sticks.

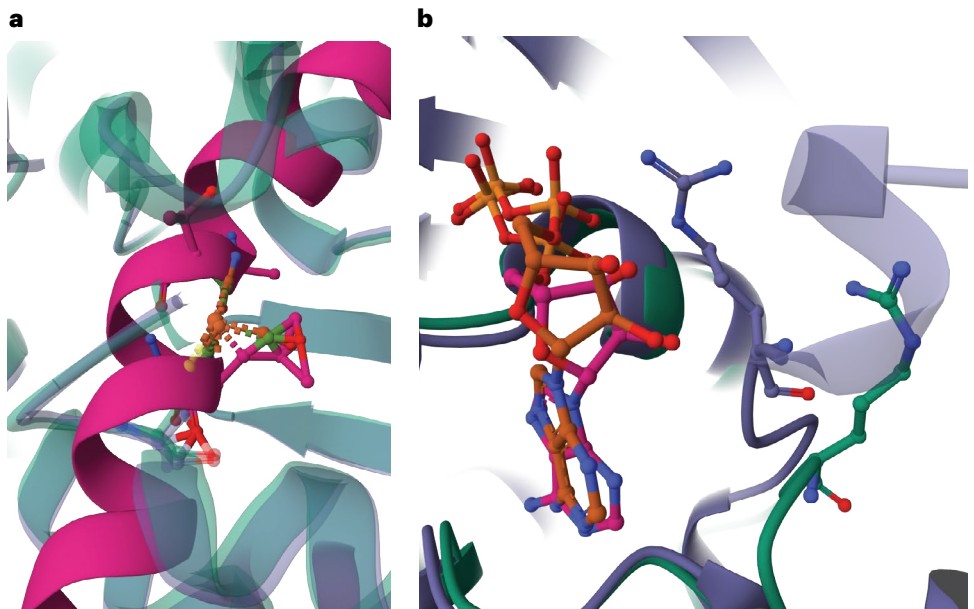

**Fig. 4 | Effect of penalizing added model contacts in LDDT-PLI. a,b**, An extra model chain in the binding site (**a**, pink) and a misplaced arginine residue (**b**), depicted with balls and sticks. Model chains are depicted in purple and reference chains in green.

and LDDT inclusion radiuses to cover the protein, and thereby allow a score to be computed. Despite this limitation, BiSyRMSD, LDDT-PLI and LDDT-LP have proven to be complementary scores showing useful properties for scoring small-molecule ligand pose predictions.

## OpenStructure as a fully automated benchmarking suite
**Overview.** The OpenStructure computational structural biology framework[49] has streamlined the implementation of comparison scores addressing various aspects discussed in this manuscript, including proteins, protein complexes and protein–ligand complexes. Furthermore,

external tools have been integrated to complement our implementations and provide a one-stop-shop for modeling method developers and benchmark assessors. The available scores are summarized in Table 1 and can be computed as described in 'Command line interface' section. A subset of scores can be invoked from a web server as described in 'Web interface' section.

**Command line interface.** The benchmarking suite is implemented in OpenStructure as executables in the form of 'actions'. Two of these actions are concerned with the comparison of theoretical models of

**Table 1 | Comparison scores in OpenStructure and their applications**

| | Protein tertiary structures | RNA tertiary structures | Protein complexes | Protein–RNA complexes | Small molecules | Chain mapping[4] | Primary citation |
|---|---|---|---|---|---|---|---|
| LDDT/bb-LDDT[1] | Yes | Yes | Yes | Yes | No | QSMap | Ref. 36 and updates in this manuscript |
| i-LDDT[1] | No | No | Yes | Yes | No | QSMap | This manuscript |
| QS-score[1] | No | No | Yes | No | No | QSMap | Ref. 28 |
| DockQ/ $f_{nat}$/ i-RMSD./ L-RMSD[2] | No | No | Yes | Yes | No | QSMap | Ref. 37 |
| ICS/IPS[2,5] | No | No | Yes | Yes | No | QSMap | Ref. 6 |
| DockQ-ave/DockQ-wave[1] | No | No | Yes | No | No | QSMap | Ref. 57 |
| GDT[2] | Yes | Yes | Yes | Yes | No | QSMapR | Ref. 33 |
| r.m.s.d.[2] | Yes | Yes | Yes | Yes | No | QSMapR | Ref. 32 |
| CAD-score[3] | Yes | Yes | Yes | Yes | No | QSMap | Ref. 35 |
| TM-score[3] | Yes | Yes | Yes | Yes | No | US-align | Ref. 25 |
| BiSyRMSD[1] | No | No | No | No | Yes | Full enumeration of binding site chains | This manuscript |
| LDDT-PLI[1] | No | No | No | No | Yes | Full enumeration of binding site chains | This manuscript |

[1]The OpenStructure scoring framework is the reference implementation. [2]Implementation in the OpenStructure scoring framework—implementation details available in Methods and comparison to reference implementation available in Supplementary Section 4. [3]External tool integrated in the OpenStructure scoring framework. [4]Only relevant for protein complexes or protein–RNA complexes, external tools either use their own chain mapping or the OpenStructure chain mapping gets injected. [5]Scores for higher order oligomers may differ from legacy implementation used until CASP15, with differences discussed in Methods.

macromolecular complexes with their reference counterpart and allow direct access to the scores described in this manuscript.

'Compare-structures' focuses on comparisons involving polymer entities, that is, protein, DNA and RNA chains.

'Compare-ligand-structures' focuses on comparisons of interactions between polymer entities and nonpolymer entities, that is, small-molecule ligands

Examples on how to run these 'actions' are available in the OpenStructure Git repository at https://git.scicore.unibas.ch/schwede/openstructure/-/blob/master/examples/scoring/README.md. This README file also includes instructions on how to set up OpenStructure using Docker, Singularity or Conda, or how to compile it from source.

**Web interface.** A selection of scores for polymer entities is also available in the SWISS-MODEL Structure Assessment server[50] available at https://swissmodel.expasy.org/assess. By providing a reference structure, users can obtain the most relevant LDDT, QS-Score, TM-Score, r.m.s.d. and DockQ scores. Automated access to the full functionality available for the command line 'actions' is available via a REST API at https://swissmodel.expasy.org/ost.

**Scoring recommendations.** In this section, we discuss common pitfalls when scoring structural predictions and provide recommendations to perform meaningful analysis in automated settings.

Structural flexibility is not taken into account in global superposition-based scores. Proteins are flexible and typically organized in relatively rigid domains whose relative orientation with respect to each other can vary[36,51]. Superposition-dependent scores fail to account for this flexibility and require manual segmentation of the reference structure[51]; in Table 1, this affects the GDT, r.m.s.d. and TM-scores. The consequences are artificially low scores that potentially overshadow accurate domain predictions. Local superposition-independent scores, such as LDDT and CAD-score, avoid this pitfall by operating on local internal contacts or distances, limiting penalties for wrongly predicted domain orientation. Therefore, LDDT and CAD-scores are generally preferable for evaluating the overall accuracy of protein structure predictions, except for the use case of differentiating relative domain orientations and overall topology.

The r.m.s.d.-based superpositions and scores can be disproportionately affected by outlier regions. GDT and TM-score are superposition dependent too but mitigate this effect by focusing on maximizing the alignment of correctly predicted regions, limiting the influence of erroneous regions by treating them as outliers. Superpositions minimizing r.m.s.d. should be applied with care or in a localized manner, with examples being BiSyRMSD or i-RMSD. L-RMSD, and consequently DockQ, are problematic as L-RMSD first derives a superposition from the full 'receptor' chain and then computes an r.m.s.d. on the full 'ligand' chain. Both steps may be affected by erroneous regions far away from any interface. The CAPRI community considers issues concerning L-RMSD by falling back on i-RMSD for these cases in their model quality classification rules[52].

Incomplete models should score lower than predictions with a complete coverage of the target sequence. Contact-based scores, along with GDT and TM-scores, naturally penalize incomplete models by design, but r.m.s.d.-based measures do not as no distance between model and reference can be computed for missing residues. From Table 1, this includes i-RMSD, L-RMSD, DockQ, r.m.s.d. and BiSyRMSD. While BiSyRMSD partially mitigates this issue by considering only complete ligand predictions, the initial superposition of the binding site can still be adversely affected by incomplete coordinates and should be carefully monitored given the provided OpenStructure output.

Incomplete reference structures, which may arise from limitations in experimental procedures, such as missing or not interpretable electron density in X-ray or cryogenic electron microscopy structures, should not result in penalties for models covering such missing regions. This affects QS-score and ICS/IPS. QS-score (referred to as QS-global in OpenStructure) is designed to compare complexes and differentiate between quaternary states. It is symmetric by design, that is, swapping the model and reference structure gives the same score. As a consequence, if the reference is incomplete, contacts that are present only in the model will penalize the score, even though the involved residues are not covered by experiment. For benchmarking scenarios that assume that the model and reference have the same stoichiometry and the model provides complete coverage, it is advisable to use the QS-best variant. This variant, available through the compare-structures action when requesting the QS-score, evaluates only the contacts between residues present in both the model and reference structure.

However, for this reason, QS-best will not penalize incomplete models or models with wrong stoichiometry. A similar situation applies to ICS/IPS. While it is beneficial to penalize contacts that exist only in the model, it is problematic if the involved residues are not covered by experimental evidence. For the benchmarking scenario of the same stoichiometry and full model coverage, the compare-structures action provides 'trimmed' variants for ICS/IPS, where the model is trimmed to include only residues that are present in the reference before score computation.

Interface centric evaluations can be conducted by i-LDDT, QS-score, DockQ/$f_{nat}$/i-RMSD/L-RMSD or ICS/IPS. The standalone CAD score executable can also perform assessment solely on interface contacts, but this feature is not integrated in the OpenStructure benchmarking suite. The DockQ family of scores assesses two-body interactions and to derive a score for higher order oligomers, DockQ-ave/DockQ-wave can be used. These two scores differ in how they aggregate per-interface contributions, with DockQ-wave weighing per-interface contributions by interface size. This can be problematic as small interfaces, which may be critical for the overall topology or biological function, get down-weighted and a simple average from DockQ-ave can be more informative. It is also important to consider that other contact based scores (i-LDDT, QS-score and ICS/IPS) can similarly be dominated by larger interfaces.

Sequence alignments are a prerequisite to establish residue-by-residue relationships between two polymer chains. All the scores in Table 1 except the TM-score use sequence-based pairwise alignments. In a benchmarking setting such as CASP or CAMEO, models are required to be numbered according to the target sequence(s). Users are advised to enforce residue number-based alignments in these cases. This has no effect on TM-score in OpenStructure as it is computed with US-align using default parameters, which performs sequence-independent alignments. It should be noted that this purely structure-based approach may result in mapping of chains with different identities.

Backbone-only scores (as in Table 1; bb-LDDT, QS-score, DockQ/i-RMSD/L-RMSD, GDT, RMSD and TM-score) only consider representative atom positions from polymer backbones and apply no penalty to incorrectly modeled side chains. Side chains are critical for protein structures, and ensuring they are properly modeled is desirable in most benchmarking scenarios.

## Discussion

Despite the large array of scores that we provide, additional use cases such as interactions between protein complexes and various molecular entities, including peptides, oligosaccharides as well as post-translational modifications, highlight further modeling challenges within this field that are still to be tackled to gain a comprehensive view of macromolecular complexes. Considerations such as structure quality validation[53] and detailed stereochemical analysis are not included in this work. Automating structural quality validation is challenging, and we are working on incorporating corresponding checks into the benchmark dataset creation process[22,54] and separately into the scores themselves. Assessment of protein–peptide interactions is currently limited by the lack of reliable alignment methods that work with arbitrary nonstandard amino acids. Finally, considerations about flexibility and disorder, involving considerably different sets of methods[55,56], are out of the scope of this manuscript.

The OpenStructure scoring framework is a comprehensive benchmarking suite providing a large array of complementary scores to assess different types of three-dimensional structure predictions in a robust and automated way. The combined scores give comprehensive deep insights into a model's accuracy. In the wake of data-driven AI-based prediction methods, high-throughput benchmarking will become increasingly critical to assess the prediction accuracy of novel methods. By providing a single, consistent and highly optimized framework,

we will facilitate future developments in the field. OpenStructure has been used extensively in CASP16 and CAMEO benchmarking efforts, proving its usefulness.

## Online content

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

## Methods

### Data input and structure preprocessing

OpenStructure accepts structural information in legacy PDB format[58] or in the preferred PDBx/mmCIF format[59]. Structures are processed as described in ref. 49. Shortly, internal residue connectivity is established on the basis of residue and atom names using the PDB chemical component dictionary (CCD)[60]. Ligands that are part of the CCD can be extracted from PDBx/mmCIF files on the basis of nonpolymer entity annotation. However, the recommended approach is to load ligand structures from SDF files[61], which inherently provide detailed connectivity information.

For polymer chain-based scores, a cleanup step replaces nonstandard residues with their corresponding parent residues as defined in the CCD (for example, SEP to SER). Residues without entry in the CCD are removed completely. In addition, hydrogen atoms, terminal OXT atoms and atoms with names that do not match the CCD are removed. Short polymer chains are excluded from scoring, with a threshold of fewer than six residues for peptides and fewer than four residues for nucleotides.

Ligand-based scores apply the same cleanup to the receptor structures, which consist of polymer chains, with the exception that nonstandard residues are not replaced. Only hydrogen atoms are removed from ligands.

### Structure visualization

Structural models in the manuscript figures were displayed in Mol*[62].

### Sequence-based grouping of polymer chains for QSMap/QSMapR

Grouping is on the basis of pairwise sequence alignments and a straightforward sequence identity measure. To minimize the likelihood of achieving high sequence identity by chance alone, QSMap/QSMapR only consider reference and model chains with a minimum length, $N$ (default of $N = 6$ for proteins and $N = 4$ for nucleotides). Pairwise sequence alignments are constructed via the Needleman–Wunsch dynamic programming algorithm[63], which employs the BLOSUM62 substitution matrix[64] for proteins and NUC44 for nucleotides (ftp://ftp.ncbi.nih.gov/blast/matrices/NUC.4.4). Alternatively, upon user request, alignments can be constructed from residue numbers, which is useful for CASP or CAMEO scenarios where participants are asked to number residues according to the submitted modeling target sequence. The grouping procedure consists of two steps: first, we need to define groups of chemically identical chains in the reference structure, and second, we need to assign each chain in the model structure to one of these groups.

By default, reference chain grouping is on the basis of sequences extracted from coordinate data. Protein and nucleotide chains are first separated, followed by clustering using a sequence identity threshold of 95%. Within each cluster, the longest chain is selected as representative. One rare failure mode arises when multiple chains originate from the same underlying target sequence but cover nonoverlapping regions, leading to their classification as different groups (for example, PDB ID 8CBU). To address this limitation, grouping information and representative sequences for reference structures are extracted from entity records when available in the PDBx/mmCIF format. If the reference structure is provided in legacy PDB format the sequence-based grouping method remains the default.

Each model chain is assigned to a reference group on the basis of its maximum sequence identity with the respective representative sequences. To allow chain mapping between homologs, the sequence identity threshold for model-to-reference assignment is relaxed to 70%. Model chains that cannot be assigned to any reference group are reported as unmapped. This threshold can be adjusted or completely removed to enforce assignment. However, doing so may lead to assignments of nonrelated chains if the model contains chains which are not represented in the reference.

### QSMap

QSMap optimizes for QS-score. The QS-score has protein-specific parameterizations, and QSMap switches to backbone LDDT (Cα for proteins and C3′ for nucleotides) with increased inclusion radius (30 Å) if nucleotides are involved. The default LDDT inclusion radius of 15 Å would be insufficient to reflect relevant pairwise distances between nucleotide backbones. While QSMap can handle protein and nucleotide chains, as well as hetero-oligomers, we describe the algorithm's complexity in terms of two equally sized homo-oligomers with $N$ chains for simplicity. QS-score and LDDT are pairwise decomposable. That is, given a model and reference complex, the overall score can be derived by separately processing contributions from individual chains and pairwise interfaces. Enumerating the full solution space of $N!$ possible mappings can be sped up by caching the computationally demanding score computations. Considering a model and reference complex with $N$ chains, there are $N(N-1)/2$ possible chain pairs in the reference ($n$ choose $k$, with $k = 2$). Given $N(N-1)/2$ possible chain pairs in the model, this results in $N(N-1)$ possible assignments for each reference chain pair (two possibilities to assign a model chain pair to a reference chain pair) and a total of $N^2(N-1)^2/2$ possible interchain contributions. Single chain contributions, which are only relevant for LDDT, add up to $N^2$. As a consequence, the score computation has a polynomial complexity of $O(N^4)$. This pushes the boundary of feasible problem sizes but still necessitates the introduction of heuristics for large $N$. We found problem sizes with $N \leq 8$ to complete with reasonable runtimes, so those can be handled by exhaustive enumeration. Larger problems are delegated to a greedy algorithm.

The greedy algorithm employs an extension strategy that starts from an initial mapping and iteratively adds pairs of model/reference chains that maximize the increase in LDDT/QS-score, as illustrated in the schematic in Extended Data Fig. 1. For efficiency, the search can be confined to pairs of chains that are 'close' or 'accessible' from the continuously updated mapping. A chain is considered accessible if its inclusion has the potential to improve the interchain component of the target score. For the QS-score (applicable to proteins only), a chain is accessible if it contains at least one representative residue position (Cβ, or Cα for glycine) within 12 Å of any chain of the same type (model or reference) already in the mapping. In addition, model or reference chains must form at least three inter-residue contacts to any chain of the same type already in the mapping. Interface contacts are defined as representative positions within 8 Å. For LDDT, a reference chain is accessible if it contains at least one representative residue position (Cα for proteins or C3′ for nucleotides) within the LDDT inclusion radius (15 Å, or 30 Å if nucleotides are involved) of any reference chain already in the mapping. For model chains, the threshold is extended to the sum of the inclusion radius and the maximum allowed distance deviation (4 Å).

In order to mitigate the risk of the algorithm being trapped in a local optimum, we sample all possible reference/model chain pairs as initial mappings ($N^2$ starting points in the case of two homo $N$-mers). In case of hetero-oligomers, all initial chain pairs must belong to the same group as defined by the algorithm described in 'Sequence-based grouping of polymer chains in QSMap/QSMapR' section. In addition we tried an approach similar to one described for US-align, where at every $n$ extension steps ($n = 3$), all possible swaps among already assigned chains were evaluated for potential score improvement to escape local optima. However, this did not improve mapping accuracy (Supplementary Fig. 2b) and was discarded. The greedy extension does not guarantee a complete mapping in the case of disconnected structures. The final algorithm enforces a full mapping.

```
For each possible reference/model chain pair:
  Use pair as initial mapping and perform greedy
  extension
  While mapping incomplete:
    For each possible unmapped reference/model chain
    pair:
      Add combination to mapping and perform greedy
      extension
      Keep mapping with highest QS-score/LDDT
  Keep complete mapping with highest QS-score/LDDT
```

## QSMapR

A multiple sequence alignment is created for each group of equivalent chains and Cα (C3′ for nucleotides) positions of columns, which are covered in each sequence, are considered for superposition and r.m.s.d. computation. To reduce runtime with limited impact on accuracy (Supplementary Fig. 2a), a subsampling by only selecting $n$ equidistant columns is performed (default $n = 50$). The same as QSMap, QSMapR samples all possible reference/model chain pairs as initial mappings ($N^2$ starting points in case of two homo $N$-mers) to start a greedy extension, as illustrated in Extended Data Fig. 1, and keeps the mapping with minimal overall r.m.s.d. In the case of hetero-oligomers, all initial chain pairs must belong to the same group, as defined by the algorithm described in 'Sequence-based grouping of polymer chains in QSMap/ QSMapR' section.

In the case of a homo $N$-mer, the greedy heuristic performs $N^3$ Kabsch minimal r.m.s.d. transformations. Naive enumeration of the solution space requires $N!$ transformations. QSMapR therefore performs naive enumeration for $N \leq 5$ ($5^3 = 125$ versus $5! = 120$) and switches to the described greedy heuristic for larger problem sizes.

## Updated LDDT reference implementation

LDDT was completely re-implemented. Extending LDDT to support protein complexes did not require conceptual changes to the algorithm described in ref. [36] and summarized in Supplementary Section 2 but involved technical adjustments to natively handle multi-chain complexes with model/reference chain mappings from QSMap/QSMapR. However, extending LDDT to nucleotides required two modifications. First, Ideal bond lengths and angles, along with their standard deviations for the stereochemistry preprocessing were expanded to include nucleotides. Amino acid and nucleotide parameters are now extracted from the CCP4 MON_LIB[65] instead of the previously used amino acid parameters from ref. [66]. No changes were made for clash checks. Second, the potential swapping of OP1/OP2 atoms in a nucleotide polymer, resulting in chemically equivalent molecules, was added to a hardcoded list of symmetries that originally only included symmetries from proteinogenic amino acids.

## Ligand definition and identification

We follow the definition of ligands from the PDB, including small molecules such as ions, solvent molecules, drugs, enzymes, co-factors, etc. associated with biological polymers[67]. In addition, the scores described here do not take biological relevance into account. This is the result of a conscious decision from our side. Other resources such as BioLiP[68], FireDB[69] or the defunct IBIS database[70] have attempted to tackle the issue. However, it is very hard to address, at least in part owing to the fact that relevance is relative and dependent on the context, and therefore is essentially impossible to automate in a general case. Therefore, we decided to focus on the assessment of ligand accuracy only. As a result, all small-molecule ligands present in the structures are assessed.

## Ligand matching and symmetry correction

We don't rely on ligand names to identify pairs of ligands in models and references, but represent all ligands by molecular graphs. If two graphs are isomorphic, they are considered a match. However, reference ligands may be incomplete owing to, for instance, missing density in the experimental structure, while model ligands are expected to be complete. Therefore, an option allows two graphs to be considered a match if the reference graph is an isomorphic subgraph of the model graph. All ligand scores described in the following sections of this manuscript can operate on incomplete reference ligands. However, post-processing must consider coverage—the fraction of model ligand atoms that are covered by the reference ligand—to avoid nonsensical matches, such as small organic solvent molecules in the reference matching organic model ligands or cofactors by pure chance. Furthermore, enumerating graph isomorphisms produces a list of one-to-one atom mappings between reference and model ligands, allowing us to account for potential symmetries that are chemically equivalent, following established methodologies[30]. Ligand atom elements serving as graph node features and graph connectivity (bonds) is established in the following order of preference: (1) loaded explicitly from an input SDF file[61], (2) extracted from the chemical component dictionary on the basis of ligand name[60] and (3) determined by a heuristic set of rules on the basis of van der Waals radii. All graph operations are performed using the Python NetworkX software[71].

## BiSyRMSD, RMSD-LP and LDDT-LP

BiSyRMSD operates on ligand atom positions and corrects for symmetries, as described in 'Ligand matching and symmetry correction' section, to report the best possible r.m.s.d. This approach is sufficient for re-docking experiments, where the reference and model polymer chains are identical and already in the same frame of reference. However, when the full model has been built in the absence of any reference information, it is necessary to appropriately transform the model ligand onto the reference ligand frame before calculating the BiSyRMSD. To mitigate the impact of structural flexibility, BiSyRMSD performs a localized transformation on the basis of the reference binding site, defined as polymer residues in close contact with the reference ligand (at least one atom within 4 Å of any reference ligand atom, excluding hydrogens). When both the reference and model consist of a single polymer chain, the corresponding binding site residues in the model are identified via sequence alignment. These residues are then used as input for a Kabsch transformation[32], utilizing the respective Cα atoms (or C3′ atoms for nucleotides), or all backbone atoms if the binding site contains two or fewer residues. For cases where the reference or model includes multiple polymer chains, mapping the model binding site becomes a chain mapping problem. The relevant set of reference polymer chains is determined by the reference binding site, whereas the relevant set of model polymer chains follows a more lenient contact definition (at least one heavy atom within 25 Å of any model ligand atom) to promote a complete mapping even with an imperfectly modeled binding site. The same sequence-based grouping used for QSMap/QSMapR ('Sequence-based grouping of macromolecule chains for QSMap/QSMapR' section) is applied to both sets. All possible mappings of model chains onto reference chains are processed. For each mapping, the model binding site residues are identified via the respective pairwise sequence alignments, and the best BiSyRMSD observed for any mapping is reported.

BiSyRMSD exclusively considers ligand atom positions. Although the binding site is critical to accommodate the ligand, it only indirectly influences BiSyRMSD through its role in the superposition process. To directly compare reference and model binding sites, RMSD-LP and LDDT-LP have been introduced, where LP stands for ligand pocket, and are similar to the previously described LDDT-BS score[72]. These methods utilize the reference/model binding site mapping obtained from BiSyRMSD to compute a backbone r.m.s.d. (using Cα atoms for proteins and C3′ atoms for nucleotides) and an all-atom LDDT, respectively.

## LDDT-PLI

LDDT-PLI is an all-atom score that, unlike BiSyRMSD, explicitly considers interactions between a ligand and its binding site. It is a

symmetry-corrected LDDT score that operates on pairwise distances between ligand and binding site with standard LDDT distance difference thresholds (0.5 Å, 1.0 Å, 2.0 Å and 4.0 Å) but with a reduced inclusion radius of 6 Å to emphasize the score's local nature. While it does not perform stereochemistry checks, LDDT-PLI is distinct in one key aspect: it considers pairwise distances within the inclusion radius in the model but not in the reference. In classical LDDT, the set of distances used for score computation is solely defined by the reference, which can be problematic for very local analyses, as intended by LDDT-PLI. For example, if a loop of a model incorrectly interacts with the ligand, the classical approach misses these interactions. LDDT-PLI addresses this by also including interatomic distances within the inclusion radius in the model, provided there is experimental evidence for both atoms involved, meaning they can be mapped to the reference. For cases where the reference or model includes multiple polymer chains, their mapping must also be considered. All chains that potentially fulfill a contact for the final LDDT-PLI score in the reference and in the model (that is, with at least one atom within the inclusion radius plus the maximum distance difference threshold of the reference ligand; 10 Å in total) are included. The same sequence-based grouping used for QSMap/QSMapR ('Sequence-based grouping of macromolecule chains for QSMap/QSMapR' section) is applied to both sets. All possible mappings of model chains onto reference chains are processed. For each mapping, model polymer residues are assigned to reference polymer residues via the respective pairwise sequence alignments. If a chain potentially contributing contacts in the model cannot be mapped to the reference at this point, the closest (by center of mass) chemically equivalent chain in the reference not already covered by the chain mapping is used, even if not initially included in the relevant set of reference polymer chains. When both the reference and model consist of a single polymer chain, the problem can be reduced to a simple pairwise sequence alignment. The optimal score is computed by simultaneously enumerating all chain mappings, all symmetries in the ligand as described in 'Ligand matching and symmetry correction' section and symmetries from the polymer chain, and the best LDDT-PLI observed for any mapping is reported.

### Ligand assignment

BiSyRMSD and LDDT-PLI are initially computed for each isomorphic pair of ligands. In this manuscript, we considered three benchmarking scenarios: (1) providing a score assessing each modeled ligand pose, (2) providing a score assessing how well each reference ligand is represented in the model and (3) providing a single score for comparing two macromolecular complexes with multiple ligands. All three scenarios require a one-to-one assignment between reference and model ligands. To be meaningful, this assignment must be nonredundant, ensuring that each model ligand is assigned to only one reference ligand and vice versa. Individual assignments are generated separately for LDDT-PLI and BiSyRMSD scores, and we found that a simple greedy algorithm yields satisfactory results. This approach iteratively assigns the best scoring pair of matching reference and model ligands until no more reference or model ligands remain to be assigned. For cases involving incomplete reference ligands, coverage—as described in 'Ligand matching and symmetry correction' section—is also considered. In each iteration, possible assignments are limited to pairs with coverage greater than the maximum coverage minus a user-specified threshold (default of 0.2). As a consequence, an assignment with a higher score is preferred, even if the coverage is slightly lower, while nonsensical assignments between small solvent ligands in the reference and large organic model ligands are discouraged.

### Implementation of external scores in OpenStructure

**DockQ, f_nat, i-RMSD and L-RMSD.** The OpenStructure implementation follows the descriptions in refs. [37,73]. The scores are designed to evaluate dimers with a defined chain assignment, eliminating the need for a chain mapping algorithm. Results from OpenStructure align closely with those from DockQ v2.1.3[74], available from https://github.com/bjornwallner/DockQ, with additional details provided in Supplementary Section 4. For protein–peptide interactions, the CAPRI community recommended modifying the default parameters[75]. This adjustment can be applied in OpenStructure by enabling the dockq-capri-peptide flag in the compare-structures 'action'.

**ICS/IPS.** The OpenStructure implementation follows the descriptions in ref. [6]. The original description does not specify procedures for chain mapping or score aggregation for higher-order oligomers. In OpenStructure, QSMap chain mapping is applied and all contacts from the complete model and reference complexes are collected to compute ICS/IPS for full complexes. Per-interface scores are reported separately. The ICS/IPS implementation used by the CASP Prediction Center (https://predictioncenter.org) aggregates per-interface scores, but it is unclear if any weighting is involved or small interfaces are discarded. Since it is not publicly available, we compared our results with those published by the Prediction Center. Results from OpenStructure closely match for dimers. Results for higher-order assemblies are qualitatively similar, with discrepancies owing to differences in score aggregation (Supplementary Section 4).

**GDT.** The GDT reports the fraction of reference Cα positions that can be superposed within a specified distance threshold of the respective model positions, which is an optimization problem with an implementation in the LGA tool[33]. OpenStructure offers its own implementation, which largely follows LGA but allows for seamless integration into the quaternary structure and RNA scoring capabilities of OpenStructure. The algorithm relies on a strict mapping between model and reference positions (Cα for peptides and C3' for nucleotides). Starting from an initial set of mapped model/reference position pairs, the following steps are iteratively applied: (1) use set to compute a minimal RMSD transformation, (2) apply the transformation to all model positions and (3) update set to include all pairs within the specified distance threshold, stopping if the set no longer changes, and report the largest set observed. Other than LGA, which employs the longest continuous segment algorithm[33] to help identify good starting sets, OpenStructure simply uses sliding windows of sizes of 7, 9, 12, 24 and 48 on the model/reference positions. To prevent long runtimes for large structures, each sliding window is applied a maximum of 1,000 times at equidistant locations.

Historically, CASP uses GDT_TS (total score), which averages GDT scores with distance thresholds 1, 2, 4 and 8 Å. The GDT_HA (high accuracy) variant, introduced in the high accuracy category of CASP7, uses lower distance thresholds of 0.5, 1, 2 and 4 Å to provide a finer-grained estimate for high-quality models[76]. OpenStructure provides the GDT_TS and GDT_HA scores, but other than LGA, which scales these scores in the range 0–100, OpenStructure scales them to 0–1. In the case of oligomers, model/reference positions are mapped on the basis of the QSMapR chain mapping algorithm. Results from OpenStructure align closely with those from LGA, as queried from https://predictioncenter.org, with additional details provided in Supplementary Section 4.

**R.m.s.d.** OpenStructure employs the Kabsch algorithm[32] to compute the r.m.s.d. on the basis of Cα positions for peptides and C3' positions for nucleotides. In the case of oligomers, chain mapping is determined by QSMapR before r.m.s.d. computation. Given that r.m.s.d. computation is a well-established procedure, no benchmarking against reference implementations was conducted.

**US-align.** OpenStructure includes US-align (version 20231222, GitHub commit 444144c) natively. Alternatively, it is possible to use a locally-installed version of US-align by supplying the path to the US-align binary.

**CAD-score.** The CAD-score is computed with the voronota-cadscore program[77], which must be installed separately from OpenStructure.

## Reporting summary

Further information on research design is available in the Nature Portfolio Reporting Summary linked to this article.

## Data availability

The analyses presented in this manuscript are based on data obtained from the CASP15 experiment (available via the Protein Structure Prediction Center at https://predictioncenter.org/download_area/CASP15/) and the RCSB PDB Search API (documented at https://search.rcsb.org/#search-api). Data and code to reproduce the figures are available via Basel University at https://git.scicore.unibas.ch/schwede/2025_scoring_paper_plots.

## Code availability

The OpenStructure source code is available at https://git.scicore.unibas.ch/schwede/openstructure under the LGPL version 3 license.

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

## Acknowledgements

We thank the sciCORE Center for Scientific Computing (https://scicore.unibas.ch) at the University of Basel for providing computational resources and system administration support. We also thank V. Oleinikovas for insightful discussions on incorporating added model contacts in LDDT-PLI. In addition, we extend our appreciation to the CASP community, with special thanks to A. Kryshtafovych for invaluable feedback on the scoring algorithms. Our gratitude goes to C. Zhang for developing a feature in US-align that allows computation of TM-scores with a predefined chain mapping. This work was supported by funding from the SIB Swiss Institute of Bioinformatics (https://www.sib.swiss/) and the Biozentrum of the University of Basel (https://www.biozentrum.unibas.ch/). J.D. and P.S. were supported by the Swiss National Science Foundation (SNSF; Ambizione grant no. 223634). The funders had no role in study design, data collection and analysis, decision to publish or preparation of the manuscript.

## Author contributions

T.S. provided conceptualization and supervision. G.S. and X.R. designed the methodology, implemented the software, performed benchmarking and validation, created visualizations and wrote the original draft of the manuscript. S.B. and G.T. contributed code, assisted with algorithm design and helped review and edit the manuscript. T.S. also participated in reviewing and editing the manuscript. J.D. and P.S. contributed to improving the scoring algorithms and provided feedback on the work. A.M.W. set up the web server for score computation. All authors read and approved the final manuscript.

## Competing interests

The authors declare no competing interests.

## Additional information

**Extended data** is available for this paper at https://doi.org/10.1038/s41592-025-02973-z.

**Correspondence and requests for materials** should be addressed to Torsten Schwede.

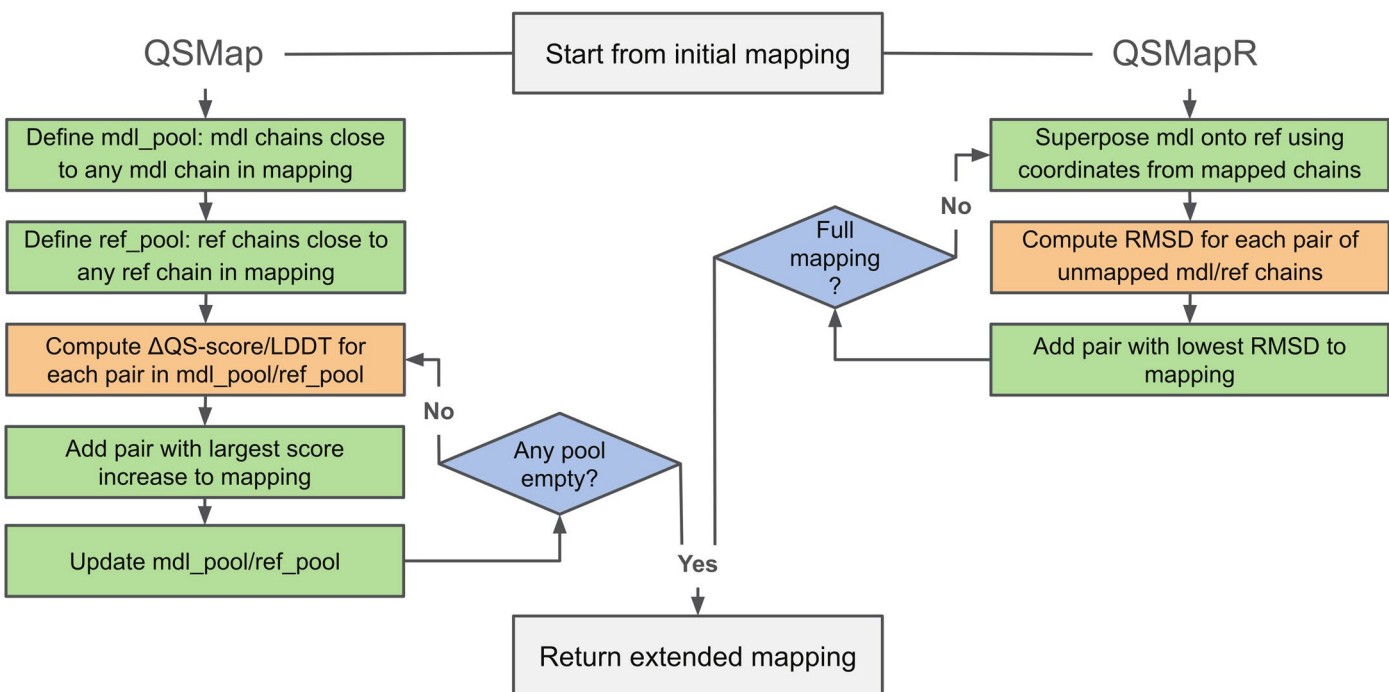

**Extended Data Fig. 1 | Greedy extension for QSMap/QSMapR.** Both algorithms extend an initial chain mapping between a model and a reference (abbreviated as mdl/ref) using a greedy strategy. Extension for QSMapR produces complete mappings, while the one for QSMap may stop early for disconnected structures, though the full QSMap algorithm still enforces a complete mapping. For hetero-oligomers, chain pairs that are considered for mapping extension (orange in workflow), must belong to the same group as defined by the algorithm described in "Sequence-based grouping of polymer chains in QSMap/QSMapR".

# Reporting Summary

## Statistics

For all statistical analyses, confirm that the following items are present in the figure legend, table legend, main text, or Methods section.

| n/a | Confirmed | |
|---|---|---|
| ☐ | ☒ | The exact sample size (*n*) for each experimental group/condition, given as a discrete number and unit of measurement |
| ☐ | ☒ | A statement on whether measurements were taken from distinct samples or whether the same sample was measured repeatedly |
| ☒ | ☐ | The statistical test(s) used AND whether they are one- or two-sided<br>*Only common tests should be described solely by name; describe more complex techniques in the Methods section.* |
| ☒ | ☐ | A description of all covariates tested |
| ☒ | ☐ | A description of any assumptions or corrections, such as tests of normality and adjustment for multiple comparisons |
| ☐ | ☒ | A full description of the statistical parameters including central tendency (e.g. means) or other basic estimates (e.g. regression coefficient) AND variation (e.g. standard deviation) or associated estimates of uncertainty (e.g. confidence intervals) |
| ☒ | ☐ | For null hypothesis testing, the test statistic (e.g. *F*, *t*, *r*) with confidence intervals, effect sizes, degrees of freedom and *P* value noted<br>*Give P values as exact values whenever suitable.* |
| ☒ | ☐ | For Bayesian analysis, information on the choice of priors and Markov chain Monte Carlo settings |
| ☒ | ☐ | For hierarchical and complex designs, identification of the appropriate level for tests and full reporting of outcomes |
| ☒ | ☐ | Estimates of effect sizes (e.g. Cohen's *d*, Pearson's *r*), indicating how they were calculated |

*Our web collection on statistics for biologists contains articles on many of the points above.*

## Software and code

Policy information about availability of computer code

| Data collection | OpenStructure (2.11.1) with Python (3.10.12), GCC (11.4.0), networkX (2.4), DockQ (2.1.3), USalign (20241108), foldseek (b4f14464bd9e3fb1921779921b985790955acd2e) were used for data collection. OpenStructure source code is available at https://git.scicore.unibas.ch/schwede/openstructure |
|---|---|
| Data analysis | Data analysis was performed with Python (3.10.12), Numpy (2.0.1), Pandas (2.2.2), Matplotlib (3.9.1), sciPY (1.14.0) and Jupyter (7.1.3). |

For manuscripts utilizing custom algorithms or software that are central to the research but not yet described in published literature, software must be made available to editors and reviewers. We strongly encourage code deposition in a community repository (e.g. GitHub). See the Nature Portfolio guidelines for submitting code & software for further information.

## Data

Policy information about availability of data

All manuscripts must include a data availability statement. This statement should provide the following information, where applicable:
- Accession codes, unique identifiers, or web links for publicly available datasets
- A description of any restrictions on data availability
- For clinical datasets or third party data, please ensure that the statement adheres to our policy

The analyses presented in this manuscript are based on data obtained from the CASP15 experiment (available at https://predictioncenter.org/download_area/

CASP15/) and the RCSB PDB Search API (documented at https://search.rcsb.org/#search-api). Data and code to reproduce the figures is available at https://git.scicore.unibas.ch/schwede/2025_scoring_paper_plots.

# Research involving human participants, their data, or biological material

Policy information about studies with human participants or human data. See also policy information about sex, gender (identity/presentation), and sexual orientation and race, ethnicity and racism.

| | |
|---|---|
| Reporting on sex and gender | n/a |
| Reporting on race, ethnicity, or other socially relevant groupings | n/a |
| Population characteristics | n/a |
| Recruitment | n/a |
| Ethics oversight | n/a |

Note that full information on the approval of the study protocol must also be provided in the manuscript.

# Field-specific reporting

Please select the one below that is the best fit for your research. If you are not sure, read the appropriate sections before making your selection.

☒ Life sciences ☐ Behavioural & social sciences ☐ Ecological, evolutionary & environmental sciences

For a reference copy of the document with all sections, see nature.com/documents/nr-reporting-summary-flat.pdf

# Life sciences study design

All studies must disclose on these points even when the disclosure is negative.

| | |
|---|---|
| Sample size | All targets and predictions provided by the modeling community for the following CASP15 categories were used: "Assembly", "Single Protein and Domain Modeling" and "Protein-ligand complexes". |
| Data exclusions | Models were excluded only if they could not be processed. No target structure was excluded. |
| Replication | Analysis has been performed on a fixed dataset. |
| Randomization | None |
| Blinding | This analysis is conducted using a publicly available dataset in which participants generated blind predictions of macromolecular structures, without prior access to the corresponding experimental ground truth. |

# Behavioural & social sciences study design

All studies must disclose on these points even when the disclosure is negative.

| | |
|---|---|
| Study description | *Briefly describe the study type including whether data are quantitative, qualitative, or mixed-methods (e.g. qualitative cross-sectional, quantitative experimental, mixed-methods case study).* |
| Research sample | *State the research sample (e.g. Harvard university undergraduates, villagers in rural India) and provide relevant demographic information (e.g. age, sex) and indicate whether the sample is representative. Provide a rationale for the study sample chosen. For studies involving existing datasets, please describe the dataset and source.* |
| Sampling strategy | *Describe the sampling procedure (e.g. random, snowball, stratified, convenience). Describe the statistical methods that were used to predetermine sample size OR if no sample-size calculation was performed, describe how sample sizes were chosen and provide a rationale for why these sample sizes are sufficient. For qualitative data, please indicate whether data saturation was considered, and what criteria were used to decide that no further sampling was needed.* |
| Data collection | *Provide details about the data collection procedure, including the instruments or devices used to record the data (e.g. pen and paper, computer, eye tracker, video or audio equipment) whether anyone was present besides the participant(s) and the researcher, and whether the researcher was blind to experimental condition and/or the study hypothesis during data collection.* |
| Timing | *Indicate the start and stop dates of data collection. If there is a gap between collection periods, state the dates for each sample cohort.* |

| Data exclusions | *If no data were excluded from the analyses, state so OR if data were excluded, provide the exact number of exclusions and the rationale behind them, indicating whether exclusion criteria were pre-established.* |
|---|---|
| Non-participation | *State how many participants dropped out/declined participation and the reason(s) given OR provide response rate OR state that no participants dropped out/declined participation.* |
| Randomization | *If participants were not allocated into experimental groups, state so OR describe how participants were allocated to groups, and if allocation was not random, describe how covariates were controlled.* |

# Ecological, evolutionary & environmental sciences study design

All studies must disclose on these points even when the disclosure is negative.

| Study description | *Briefly describe the study. For quantitative data include treatment factors and interactions, design structure (e.g. factorial, nested, hierarchical), nature and number of experimental units and replicates.* |
|---|---|
| Research sample | *Describe the research sample (e.g. a group of tagged Passer domesticus, all Stenocereus thurberi within Organ Pipe Cactus National Monument), and provide a rationale for the sample choice. When relevant, describe the organism taxa, source, sex, age range and any manipulations. State what population the sample is meant to represent when applicable. For studies involving existing datasets, describe the data and its source.* |
| Sampling strategy | *Note the sampling procedure. Describe the statistical methods that were used to predetermine sample size OR if no sample-size calculation was performed, describe how sample sizes were chosen and provide a rationale for why these sample sizes are sufficient.* |
| Data collection | *Describe the data collection procedure, including who recorded the data and how.* |
| Timing and spatial scale | *Indicate the start and stop dates of data collection, noting the frequency and periodicity of sampling and providing a rationale for these choices. If there is a gap between collection periods, state the dates for each sample cohort. Specify the spatial scale from which the data are taken* |
| Data exclusions | *If no data were excluded from the analyses, state so OR if data were excluded, describe the exclusions and the rationale behind them, indicating whether exclusion criteria were pre-established.* |
| Reproducibility | *Describe the measures taken to verify the reproducibility of experimental findings. For each experiment, note whether any attempts to repeat the experiment failed OR state that all attempts to repeat the experiment were successful.* |
| Randomization | *Describe how samples/organisms/participants were allocated into groups. If allocation was not random, describe how covariates were controlled. If this is not relevant to your study, explain why.* |
| Blinding | *Describe the extent of blinding used during data acquisition and analysis. If blinding was not possible, describe why OR explain why blinding was not relevant to your study.* |

Did the study involve field work?  ☐ Yes  ☐ No

## Field work, collection and transport

| Field conditions | *Describe the study conditions for field work, providing relevant parameters (e.g. temperature, rainfall).* |
|---|---|
| Location | *State the location of the sampling or experiment, providing relevant parameters (e.g. latitude and longitude, elevation, water depth).* |
| Access & import/export | *Describe the efforts you have made to access habitats and to collect and import/export your samples in a responsible manner and in compliance with local, national and international laws, noting any permits that were obtained (give the name of the issuing authority, the date of issue, and any identifying information).* |
| Disturbance | *Describe any disturbance caused by the study and how it was minimized.* |

# Reporting for specific materials, systems and methods

We require information from authors about some types of materials, experimental systems and methods used in many studies. Here, indicate whether each material, system or method listed is relevant to your study. If you are not sure if a list item applies to your research, read the appropriate section before selecting a response.

## Materials & experimental systems

| n/a | Involved in the study |
|-----|----------------------|
| ⊠ ☐ | Antibodies |
| ⊠ ☐ | Eukaryotic cell lines |
| ⊠ ☐ | Palaeontology and archaeology |
| ⊠ ☐ | Animals and other organisms |
| ⊠ ☐ | Clinical data |
| ⊠ ☐ | Dual use research of concern |
| ⊠ ☐ | Plants |

## Methods

| n/a | Involved in the study |
|-----|----------------------|
| ⊠ ☐ | ChIP-seq |
| ⊠ ☐ | Flow cytometry |
| ⊠ ☐ | MRI-based neuroimaging |

# Antibodies

| Antibodies used | Describe all antibodies used in the study; as applicable, provide supplier name, catalog number, clone name, and lot number. |
|---|---|
| Validation | Describe the validation of each primary antibody for the species and application, noting any validation statements on the manufacturer's website, relevant citations, antibody profiles in online databases, or data provided in the manuscript. |

# Eukaryotic cell lines

Policy information about cell lines and Sex and Gender in Research

| Cell line source(s) | State the source of each cell line used and the sex of all primary cell lines and cells derived from human participants or vertebrate models. |
|---|---|
| Authentication | Describe the authentication procedures for each cell line used OR declare that none of the cell lines used were authenticated. |
| Mycoplasma contamination | Confirm that all cell lines tested negative for mycoplasma contamination OR describe the results of the testing for mycoplasma contamination OR declare that the cell lines were not tested for mycoplasma contamination. |
| Commonly misidentified lines (See ICLAC register) | Name any commonly misidentified cell lines used in the study and provide a rationale for their use. |

# Palaeontology and Archaeology

| Specimen provenance | Provide provenance information for specimens and describe permits that were obtained for the work (including the name of the issuing authority, the date of issue, and any identifying information). Permits should encompass collection and, where applicable, export. |
|---|---|
| Specimen deposition | Indicate where the specimens have been deposited to permit free access by other researchers. |
| Dating methods | If new dates are provided, describe how they were obtained (e.g. collection, storage, sample pretreatment and measurement), where they were obtained (i.e. lab name), the calibration program and the protocol for quality assurance OR state that no new dates are provided. |

☐ Tick this box to confirm that the raw and calibrated dates are available in the paper or in Supplementary Information.

| Ethics oversight | Identify the organization(s) that approved or provided guidance on the study protocol, OR state that no ethical approval or guidance was required and explain why not. |
|---|---|

Note that full information on the approval of the study protocol must also be provided in the manuscript.

# Animals and other research organisms

Policy information about studies involving animals; ARRIVE guidelines recommended for reporting animal research, and Sex and Gender in Research

| Laboratory animals | For laboratory animals, report species, strain and age OR state that the study did not involve laboratory animals. |
|---|---|
| Wild animals | Provide details on animals observed in or captured in the field; report species and age where possible. Describe how animals were caught and transported and what happened to captive animals after the study (if killed, explain why and describe method; if released, say where and when) OR state that the study did not involve wild animals. |
| Reporting on sex | Indicate if findings apply to only one sex; describe whether sex was considered in study design, methods used for assigning sex. Provide data disaggregated for sex where this information has been collected in the source data as appropriate; provide overall |

*numbers in this Reporting Summary. Please state if this information has not been collected. Report sex-based analyses where performed, justify reasons for lack of sex-based analysis.*

Field-collected samples | *For laboratory work with field-collected samples, describe all relevant parameters such as housing, maintenance, temperature, photoperiod and end-of-experiment protocol OR state that the study did not involve samples collected from the field.*

Ethics oversight | *Identify the organization(s) that approved or provided guidance on the study protocol, OR state that no ethical approval or guidance was required and explain why not.*

Note that full information on the approval of the study protocol must also be provided in the manuscript.

# Clinical data

Policy information about clinical studies

All manuscripts should comply with the ICMJE guidelines for publication of clinical research and a completed CONSORT checklist must be included with all submissions.

Clinical trial registration | *Provide the trial registration number from ClinicalTrials.gov or an equivalent agency.*

Study protocol | *Note where the full trial protocol can be accessed OR if not available, explain why.*

Data collection | *Describe the settings and locales of data collection, noting the time periods of recruitment and data collection.*

Outcomes | *Describe how you pre-defined primary and secondary outcome measures and how you assessed these measures.*

# Dual use research of concern

Policy information about dual use research of concern

## Hazards

Could the accidental, deliberate or reckless misuse of agents or technologies generated in the work, or the application of information presented in the manuscript, pose a threat to:

No | Yes
☐ ☐ Public health
☐ ☐ National security
☐ ☐ Crops and/or livestock
☐ ☐ Ecosystems
☐ ☐ Any other significant area

## Experiments of concern

Does the work involve any of these experiments of concern:

No | Yes
☐ ☐ Demonstrate how to render a vaccine ineffective
☐ ☐ Confer resistance to therapeutically useful antibiotics or antiviral agents
☐ ☐ Enhance the virulence of a pathogen or render a nonpathogen virulent
☐ ☐ Increase transmissibility of a pathogen
☐ ☐ Alter the host range of a pathogen
☐ ☐ Enable evasion of diagnostic/detection modalities
☐ ☐ Enable the weaponization of a biological agent or toxin
☐ ☐ Any other potentially harmful combination of experiments and agents

# Plants

| Seed stocks | n/a |
|---|---|
| Novel plant genotypes | n/a |
| Authentication | n/a |

# ChIP-seq

## Data deposition

☐ Confirm that both raw and final processed data have been deposited in a public database such as GEO.

☐ Confirm that you have deposited or provided access to graph files (e.g. BED files) for the called peaks.

| Data access links
*May remain private before publication.* | *For "Initial submission" or "Revised version" documents, provide reviewer access links. For your "Final submission" document, provide a link to the deposited data.* |
|---|---|
| Files in database submission | *Provide a list of all files available in the database submission.* |
| Genome browser session
(e.g. UCSC) | *Provide a link to an anonymized genome browser session for "Initial submission" and "Revised version" documents only, to enable peer review. Write "no longer applicable" for "Final submission" documents.* |

## Methodology

| Replicates | *Describe the experimental replicates, specifying number, type and replicate agreement.* |
|---|---|
| Sequencing depth | *Describe the sequencing depth for each experiment, providing the total number of reads, uniquely mapped reads, length of reads and whether they were paired- or single-end.* |
| Antibodies | *Describe the antibodies used for the ChIP-seq experiments; as applicable, provide supplier name, catalog number, clone name, and lot number.* |
| Peak calling parameters | *Specify the command line program and parameters used for read mapping and peak calling, including the ChIP, control and index files used.* |
| Data quality | *Describe the methods used to ensure data quality in full detail, including how many peaks are at FDR 5% and above 5-fold enrichment.* |
| Software | *Describe the software used to collect and analyze the ChIP-seq data. For custom code that has been deposited into a community repository, provide accession details.* |

# Flow Cytometry

## Plots

Confirm that:

☐ The axis labels state the marker and fluorochrome used (e.g. CD4-FITC).

☐ The axis scales are clearly visible. Include numbers along axes only for bottom left plot of group (a 'group' is an analysis of identical markers).

☐ All plots are contour plots with outliers or pseudocolor plots.

☐ A numerical value for number of cells or percentage (with statistics) is provided.

## Methodology

| Sample preparation | *Describe the sample preparation, detailing the biological source of the cells and any tissue processing steps used.* |
|---|---|
| Instrument | *Identify the instrument used for data collection, specifying make and model number.* |
| Software | *Describe the software used to collect and analyze the flow cytometry data. For custom code that has been deposited into a community repository, provide accession details.* |

| | |
|---|---|
| Cell population abundance | *Describe the abundance of the relevant cell populations within post-sort fractions, providing details on the purity of the samples and how it was determined.* |
| Gating strategy | *Describe the gating strategy used for all relevant experiments, specifying the preliminary FSC/SSC gates of the starting cell population, indicating where boundaries between "positive" and "negative" staining cell populations are defined.* |

☐ Tick this box to confirm that a figure exemplifying the gating strategy is provided in the Supplementary Information.

# Magnetic resonance imaging

## Experimental design

| | |
|---|---|
| Design type | *Indicate task or resting state; event-related or block design.* |
| Design specifications | *Specify the number of blocks, trials or experimental units per session and/or subject, and specify the length of each trial or block (if trials are blocked) and interval between trials.* |
| Behavioral performance measures | *State number and/or type of variables recorded (e.g. correct button press, response time) and what statistics were used to establish that the subjects were performing the task as expected (e.g. mean, range, and/or standard deviation across subjects).* |

## Acquisition

| | |
|---|---|
| Imaging type(s) | *Specify: functional, structural, diffusion, perfusion.* |
| Field strength | *Specify in Tesla* |
| Sequence & imaging parameters | *Specify the pulse sequence type (gradient echo, spin echo, etc.), imaging type (EPI, spiral, etc.), field of view, matrix size, slice thickness, orientation and TE/TR/flip angle.* |
| Area of acquisition | *State whether a whole brain scan was used OR define the area of acquisition, describing how the region was determined.* |

Diffusion MRI    ☐ Used    ☐ Not used

## Preprocessing

| | |
|---|---|
| Preprocessing software | *Provide detail on software version and revision number and on specific parameters (model/functions, brain extraction, segmentation, smoothing kernel size, etc.).* |
| Normalization | *If data were normalized/standardized, describe the approach(es): specify linear or non-linear and define image types used for transformation OR indicate that data were not normalized and explain rationale for lack of normalization.* |
| Normalization template | *Describe the template used for normalization/transformation, specifying subject space or group standardized space (e.g. original Talairach, MNI305, ICBM152) OR indicate that the data were not normalized.* |
| Noise and artifact removal | *Describe your procedure(s) for artifact and structured noise removal, specifying motion parameters, tissue signals and physiological signals (heart rate, respiration).* |
| Volume censoring | *Define your software and/or method and criteria for volume censoring, and state the extent of such censoring.* |

## Statistical modeling & inference

| | |
|---|---|
| Model type and settings | *Specify type (mass univariate, multivariate, RSA, predictive, etc.) and describe essential details of the model at the first and second levels (e.g. fixed, random or mixed effects; drift or auto-correlation).* |
| Effect(s) tested | *Define precise effect in terms of the task or stimulus conditions instead of psychological concepts and indicate whether ANOVA or factorial designs were used.* |

Specify type of analysis:    ☐ Whole brain    ☐ ROI-based    ☐ Both

| | |
|---|---|
| Statistic type for inference<br><br>(See Eklund et al. 2016) | *Specify voxel-wise or cluster-wise and report all relevant parameters for cluster-wise methods.* |
| Correction | *Describe the type of correction and how it is obtained for multiple comparisons (e.g. FWE, FDR, permutation or Monte Carlo).* |

## Models & analysis

| n/a | Involved in the study |
|---|---|
| ☐ | ☐ Functional and/or effective connectivity |
| ☐ | ☐ Graph analysis |
| ☐ | ☐ Multivariate modeling or predictive analysis |

**Functional and/or effective connectivity**

*Report the measures of dependence used and the model details (e.g. Pearson correlation, partial correlation, mutual information).*

**Graph analysis**

*Report the dependent variable and connectivity measure, specifying weighted graph or binarized graph, subject- or group-level, and the global and/or node summaries used (e.g. clustering coefficient, efficiency, etc.).*

**Multivariate modeling and predictive analysis**

*Specify independent variables, features extraction and dimension reduction, model, training and evaluation metrics.*

