## [Peer Review File · Nature Methods]

Comparing macromolecular complexes - a fully automated benchmarking suite

Corresponding Author: Professor Torsten Schwede

Version 0:

Decision Letter:

28th Mar 2025

Dear Torsten,

Your Article, "Comparing macromolecular complexes - a fully automated benchmarking suite", has now been seen by 3 reviewers. As you will see from their comments below, although the reviewers find your work of considerable potential interest, they have raised a number of concerns. We are interested in the possibility of publishing your paper in Nature Methods, but would like to consider your response to these concerns before we reach a final decision on publication.

We therefore invite you to revise your manuscript to address these concerns. In particular, we think the manuscript could make a stronger case for novelty and utility, as well as include comparisons to existing methods and improve the writing/presentation.

Link Redacted

We hope to receive your revised paper within 6-8 weeks. If you cannot send it within this time, please let us know. In this event, we will still be happy to reconsider your paper at a later date so long as nothing similar has been accepted for publication at Nature Methods or published elsewhere.

OPEN SCIENCE REQUIREMENTS

REPORTING SUMMARY AND EDITORIAL POLICY CHECKLISTS

EXTENDED DATA FIGURES

DATA AVAILABILITY

All novel DNA and RNA sequencing data, protein sequences, genetic polymorphisms, linked genotype and phenotype data, gene expression data, macromolecular structures, and proteomics data must be deposited in a publicly accessible database, and accession codes and associated hyperlinks must be provided in the "Data Availability" section.

CODE AVAILABILITY

Please include a "Code Availability" subsection in the Online Methods which details how your custom code is made available. Only in rare cases (where code is not central to the main conclusions of the paper) is the statement "available upon request" allowed (and reasons should be specified).

For more information on our code sharing policy and requirements, please see:
<https://www.nature.com/nature-research/editorial-policies/reporting-standards#availability-of-computer-code>

MATERIALS AVAILABILITY

ORCID

Nature Methods is committed to improving transparency in authorship. As part of our efforts in this direction, we are now requesting that all authors identified as 'corresponding author' on published papers create and link their Open Researcher and Contributor Identifier (ORCID) with their account on the Manuscript Tracking System (MTS), prior to acceptance. This applies to primary research papers only. ORCID helps the scientific community achieve unambiguous attribution of all scholarly contributions. You can create and link your ORCID from the home page of the MTS by clicking on 'Modify my Springer Nature account'. For more information please visit <http://www.springernature.com/orcid>.

Sincerely,
Arunima

Arunima Singh, Ph.D.
Senior Editor
Nature Methods

Reviewers' Comments:

Reviewer #1 (Remarks to the Author):

The manuscript by Studer, Robin and co-workers presents new developments in the OpenStructure framework. Specifically, the new developments focus on improvements to metrics for comparing quaternary structures. The focus is on robust, scalable implementations for comprehensive evaluations of protein complexes and macromolecular assemblies containing protein, DNA/RNA and small molecules. These implementations are used for automatic benchmarking of methods in CASP, CAMEO and other similar assessments, that are used to benchmark structure prediction methods such as AlphaFold. In parallel, experimental methods such as single-particle cryo-EM and cryo-ET are increasingly able to characterize larger assemblies. The current developments will also be used to compare such structures. For these reasons, and given the generic applicability of the framework to compare any structures, the new developments to OpenStructure are timely and will be of broad interest to several types of biologists.

The major advances are in methods for identifying the chain mapping between the compared structures and in new ligand-based scores such as BiSyRMSD, and LDDT-PLI. Improvements to IDDT and ligand definition and identification are also presented but are relatively minor, in contrast. The authors present the technical advancements in the implementation of the metrics and also provide the command line and web interface usage along with a useful set of recommendations for a user. The following comments are on the Online Methods

1. The method for sequence-based grouping of chains and the next two methods on QSMaP/QSMaPR are not well-connected and an inexperienced reader may miss the connection between them. In particular, it is not mentioned that the chain mapping methods are for identifying a mapping between members of the same group, and these have to be applied for all groups independently. A flowchart with these methods might help.

2. Assignments occurring between non-related chains (lines 50-52): this seems to be a serious limitation. Can the authors explain why this may not be so? Is there a way this can be avoided?

3. The definition of node connectivity in the interaction graph can be made more explicit (lines 75-84). In particular, how would one know a priori if nodes can potentially contribute non-zero scores? (lines 80-81).

4. Comments on the greedy heuristic in QSMaP/QSMaPR: It is not clear why selecting accessible chains in order would lead to

better scores. This could be sensitive to the starting mapping as well as converge to a local optimum very soon. Can the authors try/discuss other heuristics tried? An alternative could be starting from the centroid chains (chains within a certain distance from the centroid) of model and reference, and incrementally adding peripheral chains.

5. The indentation in pseudocode (lines 99-105) seems to be off. Line 104 likely needs to be inside the while loop and line 105 needs to be in the for loop.

6. Minor

a. How many starting points are used in QSMaP? (lines 94-95)

b. Typo in line 97 (extra full)

c. In QSMaPR, isn't $n=50$ potentially small for large chains (line 111)?

The following comments are on the main text.

1. RMSD, TM-Score, etc are "quality measures" or "metrics for assessing structures". Perhaps these phrases should be introduced in the beginning. The word "score" in protein docking on the other hand, can also mean scores of a complex model from statistical potentials such as ZRANK. Disambiguating the above two kinds of scores would be useful.

2. FoldSeek and FoldSeek-Multimer are recent exciting advancements that allow for fast comparisons of structures. Can the authors mention these methods and put them in context of their work? Is there any complementarity in the authors' and FoldSeek's approaches and is there a plan of integrating Foldseek in future?

3. It is not clear why superposition-dependent scores are not preferred (Introduction, chain mapping, lines 67-69). Can the authors explain?

4. Can the authors briefly add to the Discussion section on whether they plan to incorporate metrics for assessing flexible regions such as protein-peptide interactions in future? What are the difficulties involved?

Minor comments

1. Line 196-197. QSMaPR is comparable to USAlign: this is not visible from the figure. Can the authors add numbers : in how many cases were the numbers similar?

2. Fig 2. What do the shades of blue or orange in panel B represent? Can they add the processor type in the legend where they are mentioning runtimes?

3. The section in Supplementary S2 is brief and not detailed enough. Can the authors elaborate more on the discrepancies between OpenStructure and PredictionCenter ICS/IPS (lines 268-271)?

Reviewer #1 (Remarks on code availability):

The github repository is well organized and the README as well as documentation is detailed enough and highly readable/usable.

The authors could provide more example commands and outputs for comparison, for the new metrics introduced in the paper. Currently only 3 commands are provided with 2 metrics (lddt, lddt-PLI)

The installation using Docker took too long and I didnt proceed to test it.

The webserver URL is not mentioned in the paper. I assumed it is <https://swissmodel.expasy.org/assess> based on what the paper says, but this does not seem to list all the latest metrics listed in the paper in Table 1. It only lists these: IDDT, QS-Score, TM-Score, RMSD, DockQ.

Reviewer #2 (Remarks to the Author):

This manuscript presents OpenStructure, a benchmarking framework designed to evaluate methods for macromolecular complex structure prediction. The framework provides a range of scoring metrics to systematically assess prediction accuracy for structures including protein-protein, protein-RNA, and protein-ligand complexes. Specifically, the authors present two newly implemented metrics: BiSyRMSD and LDDT-PLI. Additionally, the manuscript introduces a new chain-mapping algorithm, QSMaP/QSMaPR, benchmarking its computational efficiency and mapping accuracy against US-align. To illustrate the utility of BiSyRMSD and LDDT-PLI, the authors provide specific case studies demonstrating how these metrics quantify accuracy in protein-ligand interaction models. The manuscript also discusses how to choose the most suitable scoring method depending on the specific use case or type of complex being analyzed.

While OpenStructure may serve as a practical resource, particularly for researchers less familiar with common benchmarking metrics, several aspects of the manuscript fall short of demonstrating substantial scientific advancement. Primarily, the metrics provided by OpenStructure lack novelty. Although this framework is the first to integrate BiSyRMSD and LDDT-PLI into a benchmarking platform, as explicitly noted by the authors, these metrics have already been published by Robin et al. (reference numbering follows the original manuscript: [28], Assessment of protein-ligand complexes in CASP15). Furthermore, some of the key results presented in the current manuscript, such as the relationship between BiSyRMSD and LDDT-PLI

demonstrated in Figure 3, have also previously appeared in the cited reference. The notion that naive RMSD alone is insufficient for benchmarking protein-ligand interaction predictions has been established in the field. There have been other metrics to assess protein-ligand interaction predictions besides LDDT-PLI, such as those introduced by Errington et al. (2024; arXiv:2409.20227v1 [q-bio.BM]) and Morehead et al. (2024; arXiv:2405.14108 [cs.LG]). Pocket-aligned ligand RMSD has previously been employed in multiple studies, including the assessment of AlphaFold3 protein-ligand structures by Abramson et al. (2024; Nature 630, 493–500). Symmetry-corrected RMSD calculations were also employed, and there are open-source RMSD calculation tools such as `spyrmsd` (symmetry-corrected RMSD calculations in Python) by Meli and Biggin (2020; J Cheminform 12, 49). If the authors were to emphasize the novelty of their contribution, the authors should compare their newly implemented metrics against such existing methods rather than exclusively comparing the metrics to one another.

Additionally, while the proposed QSMaP/QSMaPR algorithm provides a viable alternative to existing chain mapping methods, it does not represent a substantial conceptual advance. QSMaP optimizes for QS-score, a pre-existent measure. Moreover, the manuscript fails to adequately demonstrate both the algorithm's generalizability beyond its objective function (QS-score) and its substantive advantages over US-align in the presented benchmarks.

Considering that the primary audience for this manuscript would be researchers specifically interested in method development for macromolecular structure prediction, publication in a more specialized journal dedicated to computational methodology may better align with both the manuscript's scope and the interests of its intended readership.

Reviewer #3 (Remarks to the Author):

This manuscript presents a comprehensive automated benchmarking framework for comparing predicted macromolecular models to experimental reference structures. The authors introduce novel scoring schemes (e.g., QSMaP, QSMaPR, and updated LDDT implementations) and integrate various components into the OpenStructure framework. The work represents a valuable contribution to the field and is timely given the increasing complexity of macromolecular predictions and the need for high-throughput, automated assessments. However, several key issues regarding clarity, methodological justification, and presentation remain to be addressed.

Major Concerns

1. In the Introduction, the Chain Mapping section fails to clearly articulate the challenges when dealing with identical copies of chains. This section also discusses ligand assignment under the simple heading "Chain Mapping," which is confusing. The section title should better reflect its dual focus on both chain mapping and ligand assignment, perhaps emphasizing that it aims to highlight current limitations in these approaches.
2. The manuscript criticizes RMSD for not accounting for missing atoms (Comparison Scores section under introduction). This point is questionable because, in practice, one can mask out missing residues during RMSD computation. If the intent is to penalize models with missing atoms, the authors should provide a clear explanation.
3. The manuscript states that the QS-score is more appropriate for discriminating alternative quaternary structures; however, the rationale behind this claim is not sufficiently elaborated. The explanation appears to rely solely on citations. In addition, since QSMaP and QSMaPR both utilize QS-score, the authors should include a detailed discussion of the QS-score for easier understanding.
4. In the "Mapping Polymer Chains" section under "Results", the rationale for comparing the proposed method with US-align is unclear. US-align is designed primarily for structure similarity independent of sequence, while the proposed approach leverages sequence information which can reduce mapping complexity. The authors should either justify this comparison more explicitly or include a variant of US-align that incorporates sequence data to enable a fairer comparison.
5. The manuscript does not discuss how the proposed QSMaP/QSMaPR methods compare to chain assignment algorithms used in approaches like AlphaFold-multimer or foldseek-multimer. A direct comparison or at least a discussion of the potential advantages of the proposed methods relative to these alternatives would strengthen the manuscript.
6. The results in Figure 2 showing QSMaP outperforming US-align in QS-score optimization are expected since QSMaP is specifically designed to optimize QS-score. This circularity should be acknowledged. Similarly, the improved performance on local distance-based superposition-independent scores like LDDT likely stems from the same underlying approach. Additionally, the visualization in Figure 2 would benefit from separated plots, as the overlapping data points make it difficult to distinguish between QSMaPR and US-align results for TM-score.
7. The Methods section lacks visual aids for explaining the QSMaP and QSMaPR algorithms. A diagram or schematic illustrating the chain mapping process would improve comprehension.

Minor Concerns

1. In the "Updated LDDT reference implementation" section, the phrase "as detailed elsewhere in this manuscript" (line 228) should reference the specific section where the details are provided.
2. The definition of the binding site in the "small molecule ligands" under Results section is vague. The manuscript should briefly explain whether the binding site is defined based on the ground-truth structure and specify any distance cutoffs used.

3. In Figure 3B, the locations of Asp72 and Asn77 are mentioned but not labeled. Adding clear labels would aid the reader's understanding.
4. The description spanning lines 279–282 is confusing. If the figure contains an outlier, consider differentiating its color or marker shape to improve clarity.
5. An example figure accompanying the explanation in lines 313–319 would greatly help in illustrating the described concept.
6. In Table 1, it is unclear what is meant by the "Protein" column. If it refers to protein tertiary structure, this should be explicitly stated. Moreover, marking I-LDDT as applicable despite the absence of interfaces appears inconsistent and requires clarification.
7. The sections describing the command line interface and web interface might be more appropriately placed in the supplementary material to maintain the flow of the main manuscript.

Version 1:

Decision Letter:

Our ref: NMETH-A59589A

6th Aug 2025

Dear Torsten,

Thank you for submitting your revised manuscript "Comparing macromolecular complexes - a fully automated benchmarking suite" (NMETH-A59589A). It has now been seen by the original referees and their comments are below. The reviewers find that the paper has improved in revision, and therefore we'll be happy in principle to publish it in Nature Methods, pending minor revisions to satisfy the referees' final requests and to comply with our editorial and formatting guidelines.

TRANSPARENT PEER REVIEW

ORCID

Sincerely,
Arunima

Arunima Singh, Ph.D.
Senior Editor
Nature Methods

Reviewer #1 (Remarks to the Author):

The authors have addressed all my queries.

Reviewer #2 (Remarks to the Author):

I appreciate the authors' detailed response and clarification of their contributions. However, several concerns remain for this work in its current form.

Regarding novelty and conceptual advancement:

The authors claim this work provides "novel approaches for automated scoring", but our question about the novelty of core algorithms mostly remains unanswered. While integration has value, the claim of novelty remains overstated with the main concern being about how metrics and tools inside the toolkit should be novel enough by itself.

The authors acknowledge that "the metrics themselves are not conceptually novel" and emphasizes how OpenStructure is "accessible to the community in an easy to use and robust open source package"(including BiSyRMSD, LDDT-PLI and existing metrics) along with "speed improvement (QSMAP/QSMAPR)". Apart from software engineering, the authors claim novelty in combining metrics with mapping algorithms for assessment of entire protein-ligand complex predictions with multiple chains and ligands. However, the manuscript lacks sufficient results demonstrating that OpenStructure's combination of metrics and algorithms better reflects the quality of multimeric complexes compared to existing ligand-based metrics paired with established alignment algorithms.

While the revised manuscript and response from the authors addressed some of previous concerns, the fundamental limitation remains: this work is primarily a software integration and optimization effort rather than a scientific advance.

Reviewer #3 (Remarks to the Author):

The authors have adequately addressed the majority of my previous concerns, and the manuscript is much improved. I would, however, like to suggest two minor revisions for clarity of main figures.

1. Figure 3B: While I appreciate that the labels for Asp and Asn have been added as requested, their current placement significantly overlaps with the structural elements, making them difficult to read. I suggest adjusting the label position or changing the font color to improve legibility.
2. Figure 5: The diagram uses abbreviations such as mdl and ref, which may not be immediately clear to all readers. Please consider adding a brief legend or caption clarification to explicitly define these abbreviations (e.g., mdl = model chain, ref = reference chain) to aid interpretation.

Reviewer #3 (Remarks on code availability):

Regarding the installation process, I encountered the following error when attempting to install OpenStructure via Conda as described in the GitHub repository:

Could not solve for environment specs

The following package could not be installed

└─ openstructure is not installable because it requires

└─ fftw >=3.3.10,<4.0a0 , which does not exist (perhaps a missing channel).

I was able to resolve this by explicitly installing the package using the following command:

```
conda install -c conda-forge bioconda::openstructure
```

I recommend updating the installation instructions in the GitHub repository to include this command or to clarify the required channels, as it will help users avoid confusion during setup.

Once installed, the provided example files ran successfully and reproduced the expected outputs.

Version 2:

Decision Letter:

31st Oct 2025

Dear Torsten,

I am pleased to inform you that your Article, "Comparing macromolecular complexes - a fully automated benchmarking suite", has now been accepted for publication in Nature Methods. The received and accepted dates will be January 28, 2025 and

October 31, 2025. This note is intended to let you know what to expect from us over the next month or so, and to let you know where to address any further questions.

Over the next few weeks, your paper will be copyedited to ensure that it conforms to Nature Methods style. Once your paper is typeset, you will receive an email with a link to choose the appropriate publishing options for your paper and our Author Services team will be in touch regarding any additional information that may be required. It is extremely important that you let us know now whether you will be difficult to contact over the next month. If this is the case, we ask that you send us the contact information (email, phone and fax) of someone who will be able to check the proofs and deal with any last-minute problems.

Authors may need to take specific actions to achieve compliance with funder and institutional open access mandates.

If your research is supported by a funder that requires immediate open access (e.g. according to [Plan S principles](https://www.springernature.com/gp/open-science/plan-s-compliance) or the [NIH public access policy](https://www.springernature.com/gp/open-science/us-federal-agency-compliance)) then you should select the gold OA route, and we will direct you to the compliant route where possible. Because authors warrant under our subscription licensing terms that they haven't committed to licensing any version of their article under a licence inconsistent with the terms of our agreement – including the applicable embargo period – publication under the subscription model isn't suitable for authors whose funders require no embargo.

If you are active on Twitter/X or Bluesky, please e-mail me your and your coauthors' handles so that we may tag you when the paper is published.

Best regards,
Arunima

Arunima Singh, Ph.D.
Senior Editor
Nature Methods

** Visit the Springer Nature Editorial and Publishing website at http://editorial-jobs.springernature.com?utm_source=eJP_NMeth_email&utm_medium=eJP_NMeth_email&utm_campaign=eJP_Nmeth or www.springernature.com/editorial-and-publishing-jobs for more information about our career opportunities. If you have any questions please click [here](mailto:editorial.publishing.jobs@springernature.com).

We thank the reviewers for their constructive comments and suggestions, which helped improve the manuscript. We have provided detailed responses to all points raised below.

Reviewer #1 (Remarks to the Author):

The manuscript by Studer, Robin and co-workers presents new developments in the OpenStructure framework. Specifically, the new developments focus on improvements to metrics for comparing quaternary structures. The focus is on robust, scalable implementations for comprehensive evaluations of protein complexes and macromolecular assemblies containing protein, DNA/RNA and small molecules. These implementations are used for automatic benchmarking of methods in CASP, CAMEO and other similar assessments, that are used to benchmark structure prediction methods such as AlphaFold. In parallel, experimental methods such as single-particle cryo-EM and cryo-ET are increasingly able to characterize larger assemblies. The current developments will also be used to compare such structures. For these reasons, and given the generic applicability of the framework to compare any structures, the new developments to OpenStructure are timely and will be of broad interest to several types of biologists.

The major advances are in methods for identifying the chain mapping between the compared structures and in new ligand-based scores such as BiSyRMSD, and LDDT-PLI. Improvements to IDDT and ligand definition and identification are also presented but are relatively minor, in contrast. The authors present the technical advancements in the implementation of the metrics and also provide the command line and web interface usage along with a useful set of recommendations for a user.

The following comments are on the Online Methods

1. The method for sequence-based grouping of chains and the next two methods on QSMap/QSMapR are not well-connected and an inexperienced reader may miss the connection between them. In particular, it is not mentioned that the chain mapping methods are for identifying a mapping between members of the same group, and these have to be applied for all groups independently. A flowchart with these methods might help.

We agree that the connection between the various algorithms should be obvious and easy to understand. To improve clarity, we have removed the pseudocode for both extension strategies in QSMap/QSMapR and replaced it with a schematic illustration (now included as a new Figure 5) which visually emphasizes the role of the initial grouping algorithm. Also the description on selection of starting pairs in

QSMaP/QSMaPR now clearly states that these are confined to chain pairs from the same group. We believe these changes describe the algorithms more intuitively.

2. Assignments occurring between non-related chains (lines 50-52): this seems to be a serious limitation. Can the authors explain why this may not be so? Is there a way this can be avoided?

This is a very valuable comment. The algorithm was originally designed to not only map exact matches, as encountered in benchmarking, but also to enable the comparison of homologues. During the revision we found (rare) cases in the context of CAMEO where a reference sequence that was submitted for modeling was not resolved in the reference structure that was finally deposited in the PDB. This led to exactly this behavior: Assignment of a model chain to a random reference chain. And we agree, this doesn't make much sense. Additionally, we discovered another rare failure mode in the initial grouping of reference sequences. When multiple chains originate from the same underlying target sequence but cover non-overlapping regions. Simple clustering led to their classification into separate groups. As a consequence, we have implemented changes to make the software more robust to these issues. To briefly summarize:

- Introduce sequence identity threshold of 70% when assigning model chains to groups of reference sequences. Model chains that do not fulfill this threshold for any of the reference groups are reported as unmapped. The threshold can be changed or fully omitted to restore old behavior.
- If the reference structure is provided in PDBx/mmCIF format, grouping of reference chains gets extracted from the entity records, if available. If the reference structure is provided in legacy PDB format or the entity records are missing, chains are clustered as before.

The Methods section (“Sequence based grouping of polymer chains in QSMaP/QSMaPR”) is updated accordingly. All benchmarks were recomputed to ensure that the changes do not affect the reported results.

3. The definition of node connectivity in the interaction graph can be made more explicit (lines 75-84). In particular, how would one know a priori if nodes can potentially contribute non-zero scores? (lines 80-81).

It is true that “contribution of non-zero scores” is on a very conceptual level. We rephrased the whole paragraph to make it more understandable. The new version specifies the exact criteria for a chain to potentially contribute non-zero scores which differ depending whether QS-score or LDDT is used as target score.

4. Comments on the greedy heuristic in QSMap/QSMapR: It is not clear why selecting accessible chains in order would lead to better scores. This could be sensitive to the starting mapping as well as converge to a local optimum very soon. Can the authors try/discuss other heuristics tried? An alternative could be starting from the centroid chains (chains within a certain distance from the centroid) of model and reference, and incrementally adding peripheral chains.

The concept of accessible chains is only relevant for QSMap since the target scores only consider interactions up to predefined distance thresholds. Given an incomplete mapping, we simply don't know the effect of adding a reference/model chain pair if their distance to any of the already mapped chains is beyond these thresholds. We believe that with the changes introduced for the previous comment, this becomes more clear.

The remaining challenge is convergence to local optima. During QSMap development, we explored a simple strategy, similar to one described for USalign: after every N chain assignments, all pairwise swaps among already assigned chains were evaluated for potential score improvement. However, this approach did not enhance overall chain mapping performance and was discarded. The corresponding data have been included in the supplementary Figure S2B and are referenced in the "QSMap" section of the online methods.

5. The indentation in pseudocode (lines 99-105) seems to be off. Line 104 likely needs to be inside the while loop and line 105 needs to be in the for loop.

Thank you for your feedback. We have adjusted the indentation as suggested. Additionally, we updated the pseudocode by replacing "combination" with "pair" to more clearly highlight that the algorithm adds pairs. We also made minor stylistic improvements, including the removal of dashes.

6. Minor

a. How many starting points are used in QSMap? (lines 94-95)

It's all vs all starting points. We made this more clear by replacing "In order to mitigate the risk of the algorithm being trapped in a local optimum, we use a large number of diverse starting points by using all possible reference/model chain pairs as initial mappings." with "In order to mitigate the risk of the algorithm being trapped in a local optimum, we sample all possible reference/model chain pairs as initial mappings (N^2 starting points in case of two homo N-mers). In case of hetero-oligomers, all initial chain

pairs must belong to the same group as defined by the algorithm described in “Sequence-based grouping of polymer chains in QSMap/QSMapR”

b. Typo in line 97 (extra full)

Thanks!

c. In QSMapR, isn't n=50 potentially small for large chains (line 111)?

This is a valid concern and we performed an additional analysis to measure the impact of the subsampling procedure. The respective comparison of n=50 and using all positions is available in the referenced section of the Supplementary Figure S2A. The impact on accuracy is minimal. In the text we replaced “To reduce runtime, a subsampling by only selecting n equidistant columns is performed (default: n=50)” with “To reduce runtime with minimal impact on accuracy (Figure S2A), a subsampling by only selecting n equidistant columns is performed (default n=50)”.

The following comments are on the main text.

1. RMSD, TM-Score, etc are “quality measures” or “metrics for assessing structures”. Perhaps these phrases should be introduced in the beginning. The word “score” in protein docking on the other hand, can also mean scores of a complex model from statistical potentials such as ZRANK. Disambiguating the above two kinds of scores would be useful.

Thank you for pointing this out. We have updated the introduction of the main manuscript to clarify our use of the term *score*. Specifically, we added the following to the “Comparison Scores” section in the introduction of the main manuscript:

“In this context, we use the term *score* specifically to refer to benchmarking metrics that quantify the agreement between a predicted model and a reference structure. This usage is distinct from scores that may reflect energy-based evaluations, such as those generated by tools like ZRank (Pierce and Weng 2007), which are used during modeling or docking but are not direct measures of structural similarity.”

2. FoldSeek and FoldSeek-Multimer are recent exciting advancements that allow for fast comparisons of structures. Can the authors mention these methods and put them in context of their work? Is there any complementarity in the authors' and FoldSeek's approaches and is there a plan of integrating Foldseek in future?

We are indeed living in exciting times! The growing volume of data from large protein assemblies calls for efficient benchmarking strategies and effective search tools. The latter is what Foldseek-Multimer is specifically designed for. Also in response to Reviewer 3's suggestion to put QSMapR in context of Foldseek-Multimer as well as the chain mapping algorithm described for AlphaFold-Multimer, we have implemented the AlphaFold-Multimer chain mapping algorithm within the OpenStructure package. Additionally, we expanded our benchmark to include both approaches using CASP15 data. This extended analysis is now presented in a dedicated section of the supplementary materials. Regarding Foldseek-Multimer, while it employs efficient prefiltering methods that enable fast searches across large databases, QSMapR often identifies better chain mappings in terms of TM-score with comparable runtimes when applied to actual pairs of assemblies. We conclude that for global superposition based scores, the QSMapR chain mappings are better suited for benchmarking and therefore prefer not to integrate Foldseek-Multimer.

3. It is not clear why superposition-dependent scores are not preferred (Introduction, chain mapping, lines 67-69). Can the authors explain?

Thank you for pointing this out. We agree that the original formulation was unclear. To address this, we have clarified the desired properties of a chain mapping algorithm in the introduction by adding the following:

“For robust benchmarking, the chain mapping problem is defined as the task of establishing a one-to-one assignment between chains in the model and the reference structure such that the mapping is optimal with respect to the scoring metric used to evaluate model quality. This ensures that benchmarking results reflect the best possible structural correspondence rather than artifacts of arbitrary chain assignments.”

We also emphasize in the results section that chain mapping methods operating on superposition dependent scores are suboptimal to assess a model based on superposition independent scores and vice versa. This is not a limitation of either approach, but rather a matter of applying them in the appropriate context. We have revised the text to point to the fact that a chain mapping algorithm that operates on superposition independent metrics is currently lacking in the field.

4. Can the authors briefly add to the Discussion section on whether they plan to incorporate metrics for assessing flexible regions such as protein-peptide interactions in future? What are the difficulties involved?

We have limited support for peptides with peptide specific DockQ parameters as used by CAPRI. We added the following sentence in the Online Methods section: “For protein-peptide interactions, the CAPRI community recommended modifying the default parameters (Lensink et al. 2020). This adjustment can be applied in OpenStructure by enabling the --dockq-capri-peptide flag in the compare-structures “action”.”

We do not have concrete plans to incorporate metrics for assessing protein-peptide interactions. The main challenge is alignment methods which don't work reliably for peptides, in particular those containing non-standard amino acids. We added the following sentence to the discussion section: “Assessment of protein-peptide interactions is currently limited by the lack of reliable alignment methods that work with arbitrary non-standard amino acids”.

Minor comments

1. Line 196-197. QSMapR is comparable to USAlign: this is not visible from the figure. Can the authors add numbers : in how many cases were the numbers similar?

In response to the comment regarding Foldseek-Multimer, as well as similar feedback from Reviewer 3, we expanded our benchmarking. The comparison with USalign is now available in a dedicated section of the supplementary materials. We have also improved the clarity of the comparison plots to make them more intuitive. Additionally we added annotations in the plots that report the number of cases where significant differences were observed.

2. Fig 2. What do the shades of blue or orange in panel B represent? Can they add the processor type in the legend where they are mentioning runtimes?

See comment above. The described section in the supplementary materials now contains the exact commands we used to run external tools as well as the processor type used for the computations. Additionally, we have revised the plots to improve clarity by separating them and removing the previously used blue and orange shades.

3. The section in Supplementary S2 is brief and not detailed enough. Can the authors elaborate more on the discrepancies between OpenStructure and PredictionCenter ICS/IPS (lines 268-271)?

We are aware of the discrepancies for higher order oligomers and were in contact with Andriy Kryshchak from the predictioncenter who asked us for a robust open source replacement of the previously used legacy scripts, that are not available to the community. While the legacy implementation aggregated per-interface scores to

compute a full complex score, we lack details on whether weighting was applied or if small, insignificant interfaces were discarded. As a consequence, an exact reproduction of those results was neither possible nor intended.

Instead, developed a new open source implementation from scratch to serve as a reference for the community. It strictly follows the formalism defined in Lafita et al., 2018 (PMID: 29071742) and computes the underlying precision and recall values (ICS) as well as the Jaccard coefficient of interface residues (IPS) upon aggregating all contacts across all interfaces. Additionally it also reports per-interface scores. This implementation was used in CASP16.

To inform the reader of these discrepancies we added a footnote to Table 1 where all scores are listed and updated the ICS/IPS description in the methods accordingly.

Reviewer #1 (Remarks on code availability):

The github repository is well organized and the README as well as documentation is detailed enough and highly readable/usable.

Thanks a lot!

The authors could provide more example commands and outputs for comparison, for the new metrics introduced in the paper. Currently only 3 commands are provided with 2 metrics (Iddt, Iddt-PLI)

We hope that with the improved help page (see comment below) the provided examples are concise, but sufficient. Furthermore we added pointers on how to add more scores on an opt-in basis.

The installation using Docker took too long and I didnt proceed to test it.

Thank you for your comment. While we're not certain about the exact steps you took, we recognize that running examples should be as easy as possible. We therefore significantly enhanced the examples page at https://git.scicore.unibas.ch/schwede/openstructure/-/tree/master/examples/scoring?ref_type=heads. It contains the minimal commands for setup and running scoring examples in Singularity, Docker and, as of latest, Conda. Furthermore we simplified the section "Command line interface" to only contain a brief overview of the scoring "actions" and point to the page linked above in order to give the user always the most up to date information on how to run OpenStructure.

The webserver URL is not mentioned in the paper. I assumed it is <https://swissmodel.expasy.org/assess> based on what the paper says, but this does not seem to list all the latest metrics listed in the paper in Table 1. It only lists these: IDDT, QS-Score, TM-Score, RMSD, DockQ.

Thank you for pointing this out. You are correct, the absence of the URL was misleading. We have revised the “Web interface” section to include a direct link to the structure assessment service, which provides a subset of the scores listed in Table 1 (exactly the ones you listed in your comment). Additionally, we have added information about a newly available REST API that offers the full OpenStructure scoring functionality, as defined by the scoring “actions”.

Reviewer #2 (Remarks to the Author):

This manuscript presents OpenStructure, a benchmarking framework designed to evaluate methods for macromolecular complex structure prediction. The framework provides a range of scoring metrics to systematically assess prediction accuracy for structures including protein-protein, protein-RNA, and protein-ligand complexes. Specifically, the authors present two newly implemented metrics: BiSyRMSD and LDDT-PLI. Additionally, the manuscript introduces a new chain-mapping algorithm, QSMaP/QSMaPR, benchmarking its computational efficiency and mapping accuracy against US-align. To illustrate the utility of BiSyRMSD and LDDT-PLI, the authors provide specific case studies demonstrating how these metrics quantify accuracy in protein-ligand interaction models. The manuscript also discusses how to choose the most suitable scoring method depending on the specific use case or type of complex being analyzed.

While OpenStructure may serve as a practical resource, particularly for researchers less familiar with common benchmarking metrics, several aspects of the manuscript fall short of demonstrating substantial scientific advancement. Primarily, the metrics provided by OpenStructure lack novelty. Although this framework is the first to integrate BiSyRMSD and LDDT-PLI into a benchmarking platform, as explicitly noted by the authors, these metrics have already been published by Robin et al. (reference numbering follows the original manuscript: [28], Assessment of protein-ligand complexes in CASP15). Furthermore, some of the key results presented in the current manuscript, such as the relationship between BiSyRMSD and LDDT-PLI demonstrated in Figure 3, have also previously appeared in the cited reference. The notion that naive RMSD alone is insufficient for benchmarking protein-ligand interaction predictions has been established in the field. There have been other metrics to assess protein-ligand interaction

predictions besides LDDT-PLI, such as those introduced by Errington et al. (2024; arXiv:2409.20227v1 [q-bio.BM]) and Morehead et al. (2024; arXiv:2405.14108 [cs.LG]). Pocket-aligned ligand RMSD has previously been employed in multiple studies, including the assessment of AlphaFold3 protein-ligand structures by Abramson et al. (2024; Nature 630, 493–500). Symmetry-corrected RMSD calculations were also employed, and there are open-source RMSD calculation tools such as spyrmsd (symmetry-corrected RMSD calculations in Python) by Meli and Biggin (2020; J Cheminform 12, 49). If the authors were to emphasize the novelty of their contribution, the authors should compare their newly implemented metrics against such existing methods rather than exclusively comparing the metrics to one another.

As the reviewer pointed out, the LDDT-PLI and BiSyRMSD scores have been developed by us in the context of the assessment of protein-ligand complexes in CASP15. This was done because none of the existing tools and methods was appropriate for the challenge. While it is true that the metrics themselves are not conceptually novel, we would like to emphasize that the novelty lies in how these metrics are used in combination with mapping algorithms to assess not only a single ligand docked into a crystal binding site, but rather an entire protein-ligand complex prediction, potentially consisting of multiple protein chains and ligands. In addition, while the metrics have been described before, this is the first time that they are being made accessible to the community in an easy to use, robust open source package.

Spyrmsd only calculates the symmetry-corrected RMSD of a single reference-model ligand pair in the same frame of reference and is therefore not designed for the use-cases of comparing protein-ligand complexes. It doesn't contain any logic for chain mapping, binding site detection and superposition or ligand assignment which were needed for the CASP15 evaluation. The symmetry-corrected RMSD calculations are essentially identical to ours.

Errington et al. (2024; arXiv:2409.20227v1 [q-bio.BM])'s PLIFs are in essence very similar to LDDT-PLI and evaluate how well protein-ligand interactions have been modeled. However, as we note in the paper, this type of scoring is restricted to a subjective set of interactions (for instance Errington excluded hydrophobic interactions and Van der Waals contacts) and dependent on manual preparation steps that are inapplicable on a large scale and in an automated manner.

Morehead et al. (2024; arXiv:2405.14108 [cs.LG]) employs an early version of our code to score ligands, emphasizing the need not only for a reference publication describing the scores in detail, but also for an easy to use scoring framework.

We clarified the justification for, and the novelty of the ligand scores in the introduction.

Additionally, while the proposed QSMap/QSMapR algorithm provides a viable alternative to existing chain mapping methods, it does not represent a substantial conceptual advance. QSMap optimizes for QS-score, a pre-existent measure. Moreover, the manuscript fails to adequately demonstrate both the algorithm's generalizability beyond its objective function(QS-score) and its substantive advantages over US-align in the presented benchmarks.

We agree that QSMap/QSMapR by themselves do not represent a substantial advance compared to existing methods. However, the novelty of the OpenStructure benchmarking framework is to provide robust benchmarking across diverse scoring metrics in a single, integrated robust and consistent package.

With QSMap/QSMapR, which represents the core of the OpenStructure benchmarking framework, we establish one-to-one assignment between chains in the model and the reference structure such that the mapping is optimal with respect to the scoring metric used to evaluate model quality. This enables the computation not only of TM-score, but also of many superposition-free scores, interface scores and ligand scores which are used by the community.

In addition, we show that our algorithms are about an order of magnitude faster than USalign which allows better scaling for large datasets. We also expanded the manuscript to benchmark Foldseek-Multimer and the algorithm used by AlphaFold-Multimer but found that they are not suitable for the task.

We believe that the OpenStructure benchmarking framework is a substantial improvement over existing methods, providing unique functionality not available in any other tool.

Considering that the primary audience for this manuscript would be researchers specifically interested in method development for macromolecular structure prediction, publication in a more specialized journal dedicated to computational methodology may better align with both the manuscript's scope and the interests of its intended readership.

Reviewer #3 (Remarks to the Author):

This manuscript presents a comprehensive automated benchmarking framework for comparing predicted macromolecular models to experimental reference structures. The authors introduce novel scoring schemes (e.g., QSMap, QSMapR, and updated LDDT implementations) and integrate various components into the OpenStructure framework. The work represents a valuable contribution to the field and is timely given the increasing complexity of macromolecular predictions and the need for high-throughput, automated assessments. However, several key issues regarding clarity, methodological justification, and presentation remain to be addressed.

Major Concerns

1. In the Introduction, the Chain Mapping section fails to clearly articulate the challenges when dealing with identical copies of chains. This section also discusses ligand assignment under the simple heading "Chain Mapping," which is confusing. The section title should better reflect its dual focus on both chain mapping and ligand assignment, perhaps emphasizing that it aims to highlight current limitations in these approaches.

Thank you for pointing out that this section is unclear. We changed the title to "Chemical mapping" and clarified the text to emphasize the requirement for all equivalent chemical molecules to be mapped in the use case of benchmarking. We also highlighted how existing approaches that were not designed for this use case are inappropriate and fail to produce optimal mappings with respect to the scoring metric used to evaluate model quality.

2. The manuscript criticizes RMSD for not accounting for missing atoms (Comparison Scores section under introduction). This point is questionable because, in practice, one can mask out missing residues during RMSD computation. If the intent is to penalize models with missing atoms, the authors should provide a clear explanation.

This is correct and was not formulated in a precise manner. We updated the section to emphasize the point of penalizing incomplete models by adding: "Additionally, RMSD requires subsets of mapped atom positions, meaning it does not penalize for missing residues in incomplete models and ignores any extra atoms present in one structure that are not found in the other."

3. The manuscript states that the QS-score is more appropriate for discriminating alternative quaternary structures; however, the rationale behind this claim is not sufficiently elaborated. The explanation appears to rely solely on citations. In addition,

since QSMaP and QSMaPR both utilize QS-score, the authors should include a detailed discussion of the QS-score for easier understanding.

Thanks for the comment. In fact, QS-score is only relevant for the QSMaP chain mapping algorithm. We added the section "S2 QS-score summary" in the supplementary materials. This section elaborates on the symmetry properties of QS-score, which enable the discrimination between alternative quaternary structures or stoichiometries. We reference this addition in the main text.

4. In the "Mapping Polymer Chains" section under "Results", the rationale for comparing the proposed method with US-align is unclear. US-align is designed primarily for structure similarity independent of sequence, while the proposed approach leverages sequence information which can reduce mapping complexity. The authors should either justify this comparison more explicitly or include a variant of US-align that incorporates sequence data to enable a fairer comparison.

See response to comment 6.

5. The manuscript does not discuss how the proposed QSMaP/QSMaPR methods compare to chain assignment algorithms used in approaches like AlphaFold-multimer or foldseek-multimer. A direct comparison or at least a discussion of the potential advantages of the proposed methods relative to these alternatives would strengthen the manuscript.

See response to comment 6.

6. The results in Figure 2 showing QSMaP outperforming US-align in QS-score optimization are expected since QSMaP is specifically designed to optimize QS-score. This circularity should be acknowledged. Similarly, the improved performance on local distance-based superposition-independent scores like LDDT likely stems from the same underlying approach. Additionally, the visualization in Figure 2 would benefit from separated plots, as the overlapping data points make it difficult to distinguish between QSMaPR and US-align results for TM-score.

Thank you for this comment, which highlights a core message we aim to convey: The optimal strategy for chain mapping is dependent on the specific downstream application. For example, in the context of benchmarking, using a chain mapping method that optimizes for global superposition-based metrics, such as QSMaPR or USalign, can be suboptimal when the evaluation is focused on interface centric metrics (e.g., QS-score, ICS) or superposition-independent metrics (e.g., LDDT). The inverse

also holds: methods tailored to interface or local metrics may perform poorly on global superposition dependent comparisons.

Assuming that QSMaP and USalign optimize equally well for global superposition based metrics like TM-score, it makes no difference whether we benchmark QSMaP against QSMaP or USalign. Therefore, the comparison between QSMaP and USalign primarily serves to demonstrate that QSMaP performs on par with USalign, which is widely considered to be a community gold standard.

To clarify the role of chain mapping in our work, we have revised the introduction to include the following statement:

“For robust benchmarking, the chain mapping problem is defined as the task of establishing a one-to-one assignment between chains in the model and the reference structure such that the mapping is optimal with respect to the scoring metric used to evaluate model quality. This ensures that benchmarking results reflect the best possible structural correspondence rather than artifacts of arbitrary chain assignments.”

To support this statement we applied significant changes to the results section:

- We replaced Figure 2B with a direct comparison of QSMaP and QSMaP to show the ability of QSMaP to produce better chain mappings with respect to QS-score. This change eliminates overlapping data points and improves the clarity of the plot.
- We replaced Figure 2C to compare the two methods in terms of their ability to generate an optimal chain mapping with respect to TM-score.
- We added Figure S3A in the supplementary materials to compare QSMaP and USalign to establish a comparison to the community gold standard.
- We added Figure S5A to the supplementary materials to compare the default USalign, which operates entirely sequence-independently, with USalign invoked using the `-byresi` option, which relies on residue-by-residue correspondence based on residue numbers. This partially addresses comment 4. However, the comparison remains somewhat unfair in the case of hetero-oligomers, as US-align does not perform any pre-grouping of chains while QSMaP does, which reduces complexity.
- We moved Figure 2C to the supplementary Figure S4A which depicts runtime comparisons between QSMaP, QSMaP and USalign.
- We added a comparison of QSMaP with the chain mapping algorithm utilized in AlphaFold-Multimer in the supplementary section S1. The algorithm as described

in the AlphaFold multimer paper was implemented in OpenStructure. Together with the next point, this comparison addresses comment 5.

- We added a comparison of QSMaP with the chain mapping algorithm utilized in Foldseek-Multimer. Together with the last point, this comparison addresses comment 5.

7. The Methods section lacks visual aids for explaining the QSMaP and QSMaPR algorithms. A diagram or schematic illustrating the chain mapping process would improve comprehension.

We agree that the algorithms should be described as intuitively as possible. In response to this suggestion, as well as a request from reviewer #1, we significantly reorganized the sections describing QSMaP and QSMaPR. We moved the core components which describe the greedy extension strategies to the newly added schematic illustration in Figure 5. We also better link this extension to the requirement of sequence grouping in case of hetero-oligomers. We believe that these revisions substantially improve clarity and comprehension for the reader.

Minor Concerns

1. In the "Updated LDDT reference implementation" section, the phrase "as detailed elsewhere in this manuscript" (line 228) should reference the specific section where the details are provided.

Yes, this really helps. We changed "This implementation successfully processes large assemblies by tightly integrating with the QSMaP/QSMaPR chain mapping algorithms, as detailed elsewhere in this manuscript, and was extended to nucleotides." With "This implementation successfully processes large assemblies by tightly integrating with the QSMaP/QSMaPR chain mapping algorithms, as detailed elsewhere in this manuscript (See "QSMaP" section in Methods), and was extended to nucleotides."

2. The definition of the binding site in the "small molecule ligands" under Results section is vague. The manuscript should briefly explain whether the binding site is defined based on the ground-truth structure and specify any distance cutoffs used.

The definition of binding sites is provided in detail in the Methods section. To clarify, we added the following sentence in the results section:

"The binding site is defined as any residue with at least one atom within 4Å of the ligand, excluding hydrogen atoms, based solely on the reference structure."

In addition we slightly modified the description in the Methods section as well to make it clearer.

3. In Figure 3B, the locations of Asp72 and Asn77 are mentioned but not labeled. Adding clear labels would aid the reader's understanding.

This is a very good suggestion, and we labeled the figure to make it clearer.

4. The description spanning lines 279–282 is confusing. If the figure contains an outlier, consider differentiating its color or marker shape to improve clarity.

Thank you for the suggestion. We updated the plot, figure legend and text accordingly.

5. An example figure accompanying the explanation in lines 313–319 would greatly help in illustrating the described concept.

We thank the reviewer for pointing out that this section of the manuscript was unclear.

Binding sites can only be polymer chains of sufficient length (six amino acids or four nucleotide residues, as described in the Methods section under Data input and structure preprocessing). Ligands binding to other ligands but not to polymers therefore don't have a defined binding site, and cannot be scored.

We rewrote the corresponding section to clarify this fact. We don't think this specific case warrants a separate figure in the main manuscript.

6. In Table 1, it is unclear what is meant by the "Protein" column. If it refers to protein tertiary structure, this should be explicitly stated. Moreover, marking I-LDDT as applicable despite the absence of interfaces appears inconsistent and requires clarification.

Thank you for the suggestion, we changed the Protein and RNA column titles accordingly. It is also correct that i-LDDT only operates on interfaces and we therefore moved it to a separate table row.

7. The sections describing the command line interface and web interface might be more appropriately placed in the supplementary material to maintain the flow of the main manuscript.

In response to Reviewer 1's request, we have revised both sections. The Command line interface section has been significantly shortened to present a concise overview of scoring in OpenStructure and the available installation options. Detailed usage instructions have been moved to an examples page in the Git repository, with a clear reference included in the text. Given the more streamlined nature of the revised content, we believe it is still appropriate to retain it in the main text to provide readers with a direct entry point for running the scoring functionality.

We thank the reviewers for their constructive comments and suggestions, which have helped us improve the manuscript. Below, we provide detailed responses to all points raised. Additionally, we note two changes made to ensure factual accuracy, which are reported at the end of this document.

Reviewer #1:

Remarks to the Author:

The authors have addressed all my queries.

We thank you once again for the constructive comments.

Reviewer #2:

Remarks to the Author:

I appreciate the authors' detailed response and clarification of their contributions. However, several concerns remain for this work in its current form.

Regarding novelty and conceptual advancement:

The authors claim this work provides "novel approaches for automated scoring", but our question about the novelty of core algorithms mostly remains unanswered. While integration has value, the claim of novelty remains overstated with the main concern being about how metrics and tools inside the toolkit should be novel enough by itself.

The authors acknowledge that "the metrics themselves are not conceptually novel" and emphasizes how OpenStructure is "accessible to the community in an easy to use and robust open source package"(including BiSyRMSD, LDDT-PLI and existing metrics) along with "speed improvement (QSMaP/QSMaPR)". Apart from software engineering, the authors claim novelty in combining metrics with mapping algorithms for assessment of entire protein-ligand complex predictions with multiple chains and ligands. However, the manuscript lacks sufficient results demonstrating that OpenStructure's combination of metrics and algorithms better reflects the quality of multimeric complexes compared to existing ligand-based metrics paired with established alignment algorithms.

While the revised manuscript and response from the authors addressed some of previous concerns, the fundamental limitation remains: this work is primarily a software integration and optimization effort rather than a scientific advance.

We appreciate the reviewer's attention to these important aspects and remain confident that the OpenStructure benchmarking framework represents a substantial improvement over existing approaches, offering unique functionality not available in other tools. Notably, it enabled the last two rounds of CASP assessments, which placed increased emphasis on modeling large assemblies and small molecules. This assessment would not have been possible with existing algorithms and software. Specifically, a major challenge in this context was the chain-mapping problem, which is inherently complex. The novel algorithms (QSMaP and QSMaPR) presented

in this work, seamlessly handle large macromolecular complexes, including cases where ligands interact with multiple chains, something no other software can do to the best of our knowledge. In the previous revision round, we compared OpenStructure's chain-mapping procedure to widely used alignment and chain-mapping algorithms (Supplementary Figure S3), which we found to be suboptimal. We therefore remain convinced that OpenStructure offers a more accurate and reliable solution to these problems, and constitutes a critical resource for enabling robust, reproducible and scalable benchmarking efforts to detect the next breakthroughs across the molecular modeling community.

Reviewer #3:

Remarks to the Author:

The authors have adequately addressed the majority of my previous concerns, and the manuscript is much improved. I would, however, like to suggest two minor revisions for clarity of main figures.

1. Figure 3B: While I appreciate that the labels for Asp and Asn have been added as requested, their current placement significantly overlaps with the structural elements, making them difficult to read. I suggest adjusting the label position or changing the font color to improve legibility.

We agree, label visibility has been improved.

2. Figure 5: The diagram uses abbreviations such as mdl and ref, which may not be immediately clear to all readers. Please consider adding a brief legend or caption clarification to explicitly define these abbreviations (e.g., mdl = model chain, ref = reference chain) to aid interpretation.

For consistency we now use mdl/ref throughout the figure (replaced any occurrence of model and reference) and clearly define these abbreviations in the caption.

Remarks on code availability:

Regarding the installation process, I encountered the following error when attempting to install OpenStructure via Conda as described in the GitHub repository:

Could not solve for environment specs

The following package could not be installed

└─ openstructure is not installable because it requires

└─ fftw >=3.3.10,<4.0a0 , which does not exist (perhaps a missing channel).

I was able to resolve this by explicitly installing the package using the following command:

```
conda install -c conda-forge bioconda::openstructure
```

I recommend updating the installation instructions in the GitHub repository to include this command or to clarify the required channels, as it will help users avoid confusion during setup.

Once installed, the provided example files ran successfully and reproduced the expected outputs.

Thanks for pointing out this potential pitfall! Install instructions in the Gitlab repository have been updated accordingly

Other updates:

Table 1: Footnote 1 (“Conceptually possible but never used in a benchmarking scenario”) was removed. ICS/IPS evaluated hybrid targets in CASP16 (preprint: <https://doi.org/10.1101/2025.05.29.656875>), and DockQ-related scores (f_{nat} , i-RMSD, L-RMSD) were used for protein-nucleotide complexes in the CASP16-CAPRI collaboration (preprint not available yet, personal communication, Marc Lensink from the CAPRI evaluation team).

Figure 3 caption: The LDDT-PLI default value for cases without a valid score has been corrected from -1.0 to -0.1.

Added “Author Contributions Statement” and “Competing Interests Statement” as requested by the editor.

Minor text changes

The manuscript by Studer, Robin and co-workers presents new developments in the OpenStructure framework. Specifically, the new developments focus on improvements to metrics for comparing quaternary structures. The focus is on robust, scalable implementations for comprehensive evaluations of protein complexes and macromolecular assemblies containing protein, DNA/RNA and small molecules. These implementations are used for automatic benchmarking of methods in CASP, CAMEO and other similar assessments, that are used to benchmark structure prediction methods such as AlphaFold. In parallel, experimental methods such as single-particle cryo-EM and cryo-ET are increasingly able to characterize larger assemblies. The current developments will also be used to compare such structures. For these reasons, and given the generic applicability of the framework to compare any structures, the new developments to OpenStructure are timely and will be of broad interest to several types of biologists.

The major advances are in methods for identifying the chain mapping between the compared structures and in new ligand-based scores such as BiSyRMSD, and LDDT-PLI. Improvements to IDDT and ligand definition and identification are also presented but are relatively minor, in contrast. The authors present the technical advancements in the implementation of the metrics and also provide the command line and web interface usage along with a useful set of recommendations for a user.

The following comments are on the Online Methods

1. The method for sequence-based grouping of chains and the next two methods on QSMap/QSMapR are not well-connected and an inexperienced reader may miss the connection between them. In particular, it is not mentioned that the chain mapping methods are for identifying a mapping between members of the same group, and these have to be applied for all groups independently. A flowchart with these methods might help.
2. Assignments occurring between non-related chains (lines 50-52): this seems to be a serious limitation. Can the authors explain why this may not be so? Is there a way this can be avoided?
3. The definition of node connectivity in the interaction graph can be made more explicit (lines 75-84). In particular, how would one know apriori if nodes can potentially contribute non-zero scores ? (lines 80-81).
4. Comments on the greedy heuristic in QSMap/QSMapR: It is not clear why selecting accessible chains in order would lead to better scores. This could be sensitive to the starting mapping as well as converge to a local optimum very soon. Can the authors try/discuss other heuristics tried? An alternative could be starting from the centroid chains (chains within a certain distance from the centroid) of model and reference, and incrementally adding peripheral chains.
5. The indentation in pseudocode (lines 99-105) seems to be off. Line 104 likely needs to be inside the while loop and line 105 needs to be in the for loop.
6. Minor

- a. How many starting points are used in QSMap? (lines 94-95)
- b. Typo in line 97 (extra full)
- c. In QSMapR, isn't $n=50$ potentially small for large chains (line 111)?

The following comments are on the main text.

1. RMSD, TM-Score, etc are “quality measures” or “metrics for assessing structures”. Perhaps these phrases should be introduced in the beginning. The word “score” in protein docking on the other hand, can also mean scores of a complex model from statistical potentials such as ZRANK. Disambiguating the above two kinds of scores would be useful.
2. FoldSeek and FoldSeek-Multimer are recent exciting advancements that allow for fast comparisons of structures. Can the authors mention these methods and put them in context of their work? Is there any complementarity in the authors' and FoldSeek's approaches and is there a plan of integrating Foldseek in future?
3. It is not clear why superposition-dependent scores are not preferred (Introduction, chain mapping, lines 67-69). Can the authors explain?
4. Can the authors briefly add to the Discussion section on whether they plan to incorporate metrics for assessing flexible regions such as protein-peptide interactions in future? What are the difficulties involved?

Minor comments

1. Line 196-197. QSMapR is comparable to USAlign: this is not visible from the figure. Can the authors add numbers : in how many cases were the numbers similar?
2. Fig 2. What do the shades of blue or orange in panel B represent? Can they add the processor type in the legend where they are mentioning runtimes?
3. The section in Supplementary S2 is brief and not detailed enough. Can the authors elaborate more on the discrepancies between OpenStructure and PredictionCenter ICS/IPS (lines 268-271)?

Comments on code

The github repository is well organized and the README as well as documentation is detailed enough and highly readable/usable.

The authors could provide more example commands and outputs for comparison, for the new metrics introduced in the paper. Currently only 3 commands are provided with 2 metrics (Iddt, Iddt-PLI)

The installation using Docker took too long and I didnt proceed to test it.

The webservice URL is not mentioned in the paper. I assumed it is <https://swissmodel.expasy.org/assess> based on what the paper says, but this does not seem to list all the latest metrics listed in the paper in Table 1. It only lists these: IDDT, QS-Score, TM-Score, RMSD, DockQ.